# Waste to Medicine: Evidence from Computational Studies on the Modulatory Role of Corn Silk on the Therapeutic Targets Implicated in Type 2 Diabetes Mellitus

**DOI:** 10.3390/biology12121509

**Published:** 2023-12-11

**Authors:** Ayesha Akoonjee, Adedayo Ayodeji Lanrewaju, Fatai Oladunni Balogun, Nokwanda Pearl Makunga, Saheed Sabiu

**Affiliations:** 1Department of Biotechnology and Food Science, Faculty of Applied Sciences, Durban University of Technology, Durban 4000, South Africa; 22290008@dut4life.ac.za (A.A.); 22172511@dut4life.ac.za (A.A.L.); fataib@dut.ac.za (F.O.B.); 2Department of Botany and Zoology, Stellenbosch University, Private Bag X1, Matieland, Stellenbosch 7602, South Africa; makunga@sun.ac.za

**Keywords:** *ADORA1*, cAMP signaling pathway, chromatography, corn silk, *GABBR1*, *HCAR2*, in silico techniques, systems biology, African traditional medicine, type 2 diabetes mellitus

## Abstract

**Simple Summary:**

Type 2 diabetes mellitus (T2DM) is characterized by insulin resistance and/or defective insulin production in the human body. Corn silk (CS), an abundant, readily available and affordable waste product of corn cultivation, has extensive therapeutic applications against various diseases including T2DM. Although the antidiabetic potential of CS is well-established, the understanding of the mechanism of action (MoA) behind its reported antidiabetic potentials is lacking. Hence, determining its MoA may provide laudable insight contributing towards developing an effective drug candidate for combating the ill effects of T2DM.

**Abstract:**

Type 2 diabetes mellitus (T2DM) is characterized by insulin resistance and/or defective insulin production in the human body. Although the antidiabetic action of corn silk (CS) is well-established, the understanding of the mechanism of action (MoA) behind this potential is lacking. Hence, this study aimed to elucidate the MoA in different samples (raw and three extracts: aqueous, hydro-ethanolic, and ethanolic) as a therapeutic agent for the management of T2DM using metabolomic profiling and computational techniques. Ultra-performance liquid chromatography-mass spectrometry (UP-LCMS), in silico techniques, and density functional theory were used for compound identification and to predict the MoA. A total of 110 out of the 128 identified secondary metabolites passed the Lipinski’s rule of five. The Kyoto Encyclopaedia of Genes and Genomes pathway enrichment analysis revealed the cAMP pathway as the hub signaling pathway, in which *ADORA1*, *HCAR2*, and *GABBR1* were identified as the key target genes implicated in the pathway. Since gallicynoic acid (−48.74 kcal/mol), dodecanedioc acid (−34.53 kcal/mol), and tetradecanedioc acid (−36.80 kcal/mol) interacted well with *ADORA1*, *HCAR2*, and *GABBR1*, respectively, and are thermodynamically stable in their formed compatible complexes, according to the post-molecular dynamics simulation results, they are suggested as potential drug candidates for T2DM therapy via the maintenance of normal glucose homeostasis and pancreatic β-cell function.

## 1. Introduction

Diabetes mellitus (DM), a chronic, uncurable metabolic disease characterized by elevated blood glucose levels, is a significant contributor to morbidity, mortality, and health costs worldwide [1]. In 2021, an estimated 24 million adults (20–79 years old) in Africa were living with DM and approximately 13 million of these adults remain undiagnosed [2]. The International Diabetes Federation (IDF) stipulated that, in 2021, around 4,234,000 South African adults had DM out of a total adult population of 37,416,800, indicating a prevalence of 1 in 11 adults [3]. Type 2 diabetes mellitus (T2DM) is the most prevalent type of diabetes, constituting roughly 90% of all DM cases [4], and, if left untreated, can result in severe complications, including kidney disease, ocular damage, cardiovascular diseases (e.g., stroke and heart attack), increased risk of bone fracture, nerve damage, skin conditions, and hearing impairment [5]. The pathogenesis of T2DM lies in defective insulin production by β-cells and/or insulin resistance in insulin-sensitive cells [5]. Although the efficacy of many synthetic hypoglycaemic drugs such as sulfonyl ureas, biguanides, alpha-glucosidase inhibitors, thiazolidinediones, and non-sulfonyl urea secretagogues are undoubtedly potent in the management of T2DM [6], their limitations, including long-term regimens, high cost, efficacy and individual variability as well as the elicitation of considerable side effects (nausea, weight gain or loss, cardiovascular complications, and gastrointestinal discomforts) have undermined their application in clinical practice [7]. Consequently, cost-effective therapeutic agents with significant antidiabetic activity and little or no toxic effects are highly sought out as alternative T2DM therapeutics [8].

Medicinal plants (MPs) have been utilized in traditional medicine systems for centuries, offering a wealth of knowledge and therapeutic possibilities for the treatment of various ailments, including T2DM [7]. Currently, over 1200 species of plants have been traditionally used as natural antidiabetics globally, many of which are being explored for their potential hypoglycaemic properties [8]. For example, corn silk (CS) is an abundant waste plant material of corn, *Zea mays* (L.) which has been used in traditional medicine for several applications, including as a potential remedy for T2DM [9,10]. Corn silk is described as pale green, yellow, or light brown thread-like strands, crucial for the successful pollination of corn kernels [11]. A considerable number of phytochemicals have been identified in CS, including phenolic acids, flavonoids, carotenoids, tannins, sterols, volatile compounds, sugars, vitamins, minerals, polysaccharides, proteins, and peptides responsible for a diverse range of promising pharmacological properties not limited to antioxidant, anti-hyperlipidemic, antibacterial, anti-cancer, antihypertensive, antidiabetic, diuretic, and kaliuretic [7,8,12], making CS a valuable natural resource for healthcare applications [12,13]. Although numerous studies have reported the promising antidiabetic action of CS [11,14,15,16,17,18,19], additional research is needed to establish the mechanism of antidiabetic action for it to be considered as a possible therapeutic agent in the management of T2DM [20].

With the rapid development in technology, the field of drug discovery has been revolutionized, accelerating the discovery and development of therapeutic drugs [21]. This expands the possibilities of quickly identifying, designing, optimizing, and developing new therapeutic agents [22]. The use of computational techniques such as network pharmacology (NP), molecular docking, and molecular dynamics (MD) simulation is essential in drug discovery, allowing for the understanding of interactions between drugs and their target proteins at different biological levels [23]. Network pharmacology (NP) integrates systems biology, network science, and pharmacology to provide a comprehensive understanding of the mechanisms of drug action, the prediction of drug targets, the pathogenesis of diseases, and the optimization of therapeutic outcomes [24]. The interaction between drug targets and multiple components such as proteins, genes, and signaling pathways in a biological system are better explored using NP analysis [25], providing a unique perspective on drug action and disease mechanisms by considering the complexity of biological systems in order to provide insights into drug development and optimization [26]. Interestingly, there have been numerous studies that have investigated the antidiabetic mechanism of action (MoA) of natural products, plants, or plant fractions using NP [19,27,28,29,30]. Molecular docking is a cost-effective, fast, and reliable computational method, employed in drug discovery, medicinal chemistry, and structural biology, to examine the interactions between a small molecule (ligand) and a target biomolecule, typically a protein [31]. Natural products, plants, and plant fractions may be assessed against different targets implicated in disease emergence (and in this case, T2DM), such as key enzymes and genes, thus allowing for a comprehensive understanding of the antidiabetic potentials of CS typically against these targets through this approach [23,29]. Molecular dynamics (MD) simulation on the other hand, being a powerful computational technique, is used to the study the behaviour and interactions of atoms and molecules over time [32,33]. The approach similarly provides insight into protein–ligand interactions and the MoA of therapeutic agents, which can prove useful in the discovery and development of therapeutic agents, particularly T2DM [33,34,35,36]. The integration of various technologies, particularly NP, molecular docking, and MD simulation, as adopted in this study, should provide information towards identifying novel therapeutic agents from CS, which can be discovered and/or developed as alternative agents for the management of T2DM [37]. The present study is thus aimed at studying the antidiabetic MoA of secondary metabolites in various extracts of CS, with key targets implicated in T2DM through metabolomic profiling, NP, molecular docking, and MD simulation for the discovery and development of novel therapeutic agents for the management of T2DM.

## 2. Materials and Methods

### 2.1. Silk Collection, Processing and Extract Preparation

Fresh CS of the commercial hybrid ILHYB22, a commonly consumed cultivar in South Africa, was harvested at the Cedara College of Agriculture in KwaZulu Natal, South Africa. The CS was washed with water to remove dirt and other contaminants before being air-dried for three days [38]. The dried CS was then powdered to a constant weight using an electric grinder (SM-450, Mills, MRC Laboratory Instruments, Twickenham, UK) [39]. The powdered materials were used for the preparation of extracts (aqueous, hydro-ethanol, and ethanol). Briefly, for the preparation of the aqueous extract, 150 g of CS powder was boiled at 100 °C in 1.5 L distilled water for 30 min [39,40], followed by filtration (Whatman No.1 filter paper) and lyophilization (Telstar Lyoquest Arctic, Tokyo, Japan) [10]. The hydro-ethanol and ethanol extracts were prepared by macerating approximately 100 g each of CS powder in 1 L of 50% ethanol and absolute ethanol, respectively, at 150 rpm, using an orbital shaker (Labnet Orbit LS, Edison, NJ, USA) for three days, followed by filtration (Whatman No.1 filter paper) [39]. The filtrates were concentrated using a rotary evaporator (HEI-VAP Core, 571-01310-00, Heidolph, Schwabach, Germany) and the leftover water from the hydro-ethanol filtrate was subsequently lyophilized to complete the extract preparation [39]. All the extracts were refrigerated at 4 °C until needed [10].

### 2.2. Ultra-Performance Liquid Chromatography-Mass Spectrometry Analysis

Four samples of CS [raw and three extracts (aqueous, hydro-ethanolic, and ethanolic)] were used for identification of the phytoconstituents present in CS through ultra-performance liquid chromatography-mass spectrometry (UPLC-MS) analysis according to the methods of Mangana et al. [41] and Mangana et al. [42]. Analysis of the CS samples was performed by utilizing a Water Synapt G2 quadruple time-of-flight mass spectrometer which was connected to a Waters Acquity UPLC-combined photo diode array detector (Milford, Massachusetts, United States of America). Extraction of the sample was performed with 2 g of each sample by employing a solvent system containing 50% methanol and 0.1% formic acid for 24 h at room temperature. The samples were then vortexed (model VX-200 S0200, Labnet, Edison, NJ, USA) for 1 min and extraction was performed using an ultrasonic bath (SS-6508T, Sunshine Scientific Equipment, Delhi, India) for 1 h. A volume of 1 mL of the residue was then withdrawn and centrifuged (mySPIN 12, Thermo Scientific, Waltham, MA, USA) at 14,000× *g* rpm 5 min. Ionization was accomplished using an electrospray source using a cone voltage of 15 V and capillary voltage of 2.5 kV, wherein only the negative mode was employed. Nitrogen was employed as the desolvation gas at 650 L h^−1^ and the desolvation temperature was set to 275 °C. A Waters UPLC BEH C18 column (2.1 × 100 mm^2^, 1.7 µm particle size) was utilized and 2 µL was injected for analysis. The gradient commenced at 95%, consisting of 0.1% (*v*/*v*) formic acid (solvent A) and 5% acetonitrile (solvent B). This was followed by a gradient of 60%, consisting of 0.1% solvent A and 40% solvent B at 9 min; 30% solvent A and 70% solvent B over 9.1 min; 100% solvent B at 14 min; and 95% solvent A and 5% solvent B at 14.01 min. The conditions thereafter remained constant to a total run time of 15 min, with solvent A at 95% and solvent B at 5%. Acquisition of data was performed through employment of MassLynx4.1 software. The detection and confirmation of compounds were processed using the MS-DIAL and MS-FINDER software 2.0 (RIKEN Center for Sustainable Resource Science: Metabolome Informatics Research Team, Kanagawa, Japan). Principal component analysis (PCA) scores plot was applied as previously discussed by [42] using the database Metaboanalyst (https://www.metaboanalyst.ca/MetaboAnalyst/) (accessed on 1 July 2022).

### 2.3. Network Pharmacology

#### 2.3.1. Pharmacokinetic Properties of Corn Silk Phytoconstituents

The pharmacokinetic properties of the UPLC-MS identified secondary metabolites of CS were evaluated using Lipinski’s rule of five (Ro5) on the SwissADME server (http://www.swissadme.ch/; accessed on 30 July 2022) to predict their drug-likeness property [25]. The Simplified Molecular Input Line Entry System (SMILES) of the CS secondary metabolites were obtained from PubChem website (https://pubchem.ncbi.nlm.nih.gov/; accessed on 30 July 2022) to identify orally bioavailable compounds [25]. Compounds with 2 or less violations (<5 hydrogen bond donors; ≤10 hydrogen bond acceptors; molecular weight ≤ 500 g/mol and partition co-efficient < 5) were considered to pass the pharmacokinetics analysis [43].

#### 2.3.2. Acquisition of CS Phytoconstituents and T2DM-Associated Targets

The acquisition of therapeutic targets related to phytoconstituents of CS and T2DM was achieved as previously reported by Akoonjee et al. [29]. Identification of target genes related to CS secondary metabolites was performed through employing both Swiss Target Prediction (STP) (http://www.swisstargetprediction.ch/; accessed on 1 August 2022) and Similarity Ensemble Approach (SEA) (https://sea.bkslab.org/; accessed on 1 August 2022) databases to avoid biases, while the T2DM-related target genes were identified from GeneCards database (https://www.genecards.org/; accessed on 1 August 2022). Subsequently, Venny 2.1 (https://bioinfogp.cnb.csic.es/tools/venny/; accessed on 10 August 2022) was employed to identify and characterize the overlapping targets between CS secondary metabolites and T2DM target genes [30].

#### 2.3.3. Protein–Protein Interaction Network Construction and Analyses of KEGG Enrichment Pathway, Gene Ontology, and Compound–Target Pathway of Overlapping Target Genes

The Search Tool for the Retrieval of Interacting Genes/Proteins (STRING) database (https://string-db.org/; accessed on 1 September 2022) was utilized to correlate the network analysis of the overlapping T2DM target genes related to CS secondary metabolites and Kyoto Encyclopaedia of Genes and Genomes (KEGG) pathway enrichment analysis to identify key T2DM-related signaling pathways associated with the overlapping genes [28]. The Database for Annotation, Visualization and Integrated Discovery (DAVID) (https://david.ncifcrf.gov/tools.jsp; accessed on 11 November 2022) was adopted to execute gene ontology analysis related to this study, while the software Cytoscape 3.6.0 with the built-in merger algorithm was employed to correlate and visualize compound–target network pathways and gene–compound interaction networks [29].

### 2.4. Molecular Docking and MD Simulation of T2DM-Related Target Genes with CS Secondary Metabolites

Following NP analysis, the key CS secondary metabolites and targets identified were subjected to molecular docking, as previously reported [44,45]. Briefly, the X-ray crystal structures of the identified key T2DM-related target genes such as adenosine receptor A1 (*ADORA1*) (PDB ID: 5UEN) and gamma-aminobutyric acid type B receptor subunit (*GABBR1*) (PDB ID: 4MQF) were obtained from RSCB Protein Data Bank (https://www.rcsb.org/; accessed on 1 September 2022), while the X-ray crystal structure of the target gene hydroxycarboxylic acid receptor 2 (*HCAR2*) (AlphaFold ID: Q8TDS4) was obtained from AlphaFold protein structural database (https://alphafold.ebi.ac.uk/; accessed on 1 September 2022). Preparation of the T2DM-related target genes was performed using USCF Chimera v 1.16 through the removal of water molecules and protein residue connectivity [44]. Three-dimensional (3D) conformers of the CS secondary metabolites as well as the reference standards metformin and resveratrol were obtained from PubChem (https://pubchem.ncbi.nlm.nih.gov/; accessed on 1 September 2022) as previously reported [30]. The addition of Gasteiger charges and non-polar hydrogen atoms was performed on USCF Chimera v 1.16 for optimization of the 3D conformers [45]. Thereafter, the optimized secondary metabolites were individually docked at the active site of their respective T2DM-related target genes (*ADORA1*, *HCAR2*, and *GABBR1*) using Autodock Vina Plugin on Chimera v 1.16 [44]. The docking of the CS phytoconstituents at the active sites of the therapeutic targets was performed through the adjustment of the grid box coordinates to match the established x, y, and z coordinates of the native ligand obtained with Discovery Studio version 21.1.0. Docking protocol validation was carried out to prevent pseudo-positive binding conformations [45] by measuring root mean square deviation (RMSD) of docked ligands from the reference pocket bearing the native ligands in the experimental co-crystal structures of *ADORA1* (Figure 1a), *HCAR2* (Figure 1b), and *GABBR1* (Figure 1c), following optimal superimposition [45]. The RMSD values (0.5 Å) obtained between the docked ligands from the native inhibitor in the 3D structures of *ADORA1*, *HCAR2*, and *GABBR1* indicated the same binding orientation, ultimately validating the protocol adopted [45]. Based on the docking scores, the five complexes with the best pose (most negative docking score) against each target was selected for further analysis through a 120 ns MD simulation, as detailed by Sabiu et al. [46].

The GPU version with the AMBER 18 package (Centre for High Performance and Computing) system with the FF18SB variant of the AMBER force field was used for the MD simulation analysis. To generate atomic partial charges for the CS compounds, ANTECHAMBER was utilized, employing the restrained electrostatic potential (RESP) and the general amber force field (GAFF) procedures. The Leap module of AMBER 18 was used to add hydrogen atoms and Na+ and Cl- counterions to neutralize all systems. These systems were then placed in orthorhombic boxes filled with TIP3P water molecules, ensuring that all atoms were within 8 Å of any box edge. An initial minimization of 2000 steps was conducted, applying a restraint potential of 500 kcal/mol to both solutes. This minimization included 1000 steps using the steepest descent method, followed by 1000 steps using the conjugate gradients method. Subsequently, a full minimization of 1000 steps was performed using the conjugate gradient algorithm without any restraint. Gradual heating from 0 K to 300 K was carried out for 50 ps, ensuring all systems maintained a consistent number of atoms and volume. During this process, a potential harmonic restraint of 10 kcal/mol and a collision frequency of 1.0 ps were applied to the solutes within the systems. Following heating, an equilibration estimating 500 ps of each system was carried out wherein the operating temperature and pressure were kept consistent at 300 K and 1 bar, respectively, for simulation of an isobaric–isothermal ensemble [46]. Furthermore, additional features, such as several atoms and the pressure were kept constant, mimicking an isobaric–isothermal ensemble. The system’s pressure was maintained at 1 bar, employing the Berendsen’s barostat, while the MD simulation lasted for 120 ns. In each simulation, the SHAKE algorithm was employed to constrict the bonds of the hydrogen atoms. The step size of each simulation was 2 fs and an SPFP precision model was used. The simulations coincided with the isobaric–isothermal ensemble (NPT), with randomized seeding, the constant pressure of 1 bar maintained by a pressure-coupling constant of 2 ps, a temperature of 300 K, and Langevin thermostat with a collision frequency of 1.0 ps [46]. The post-dynamics data were examined as previously described by Aribisala et al. [44]. Post-MDS parameters such as root mean square deviation (RMSD), root mean square fluctuation (RMSF), radius of gyration (RoG), solvent accessible surface (SASA), and number of hydrogen bonds were investigated using the CPPTRAJ module incorporated in the AMBER 18 suite. The free binding energy of the formed complexes were calculated using the Molecular Mechanics/GB Surface Area (MM/GBSA) method, as detailed by Sabiu et al. [46]. The average binding free energy was calculated over 100,000 snapshots obtained from the 120 ns trajectory. The free binding energy (ΔG) for each molecular species (complex, ligand, and protein) was calculated using the expressions [(1)–(5)] below:Δ*G_bind_* = *E_gas_* + *G_sol_* − *TS*(1)
Δ*G_bind_* = *G_complex_* − *G_receptor_* − *G_ligand_*(2)
*E_gas_* = *E_int_* + *E_vdw_* + *E_ele_*(3)
*G_sol_* = *G_GB_* + *G_SA_*(4)
*G_SA_* = *γ**SASA*(5)

The ligand–receptor complexes’ interaction at the active sites in each treatment case was identified with post-MDS and visualized using Discovery Studio version 21.1.0 [44].

### 2.5. Quantum Chemical Calculations

The electronic properties of the compounds were investigated using the density functional theory (DFT) method available in the Gaussian 16 suite while Gauss View v 6.0 to view the output files [47]. The functional Becke3-Lee–Yang–Parr (B3LYP) method combined with the 6–31 + G(d,p) basis set was employed for the geometry optimization of the compounds [47]. The study assessed the conceptual DFT (cDFT); namely, the energies of the lowest unoccupied molecular orbital (E_LUMO_) and the highest occupied molecular orbital (E_HOMO_). Thereafter, other chemical descriptors such as energy gap (Δ*E*), ionization energy (I), electron affinity (A), chemical hardness (ŋ), softness (δ), electronegativity (χ), chemical potential (Cp), global electrophilicity (Ꞷ) were calculated using the equations below, as described previously [47].
Δ*E* = E_LUMO_ − E_HOMO_(6)
I = −E_LUMO_(7)
A = −E_HOMO_(8)
*ŋ* = (ΔE)/2(9)
δ = 1/(ŋ)(10)
χ = (I + A)/2(11)
Cp = −χ(12)
Ꞷ = χ^2^/(Δ*E*)(13)

## 3. Results

### 3.1. Metabolomic Profiling

#### 3.1.1. Ultra-Performance Liquid Chromatography-Mass Spectrometry

The data obtained with respect to the 128 identified phytoconstituents from the metabolites of the investigated CS extracts by UPLC-MS analysis is presented in Appendix A and was confirmed on the chromatograms produced from MassLynx (Appendix A).

#### 3.1.2. Principal Component Analysis

The principal component analysis results, which indicated the presence of differences (qualitative and quantitative) between the various samples of the CS, are presented in Appendix A. The highest amount of variance between the raw CS and various extracts (aqueous, hydro-ethanolic, and ethanolic) was 67% (46.8% observed in principal component 1 and 20.2% between the samples in principal component 2). Among the samples investigated, the aqueous, hydro-ethanol, and raw CS samples showed more similarity in the chemical diversity of the phytoconstituents in comparison to the ethanolic CS. There were variations in the amount of secondary metabolites present in the different samples of CS, as observed in Appendix A.

### 3.2. Drug Candidate Filtering/ADME Property Analysis

A total number of 110 phytocompounds from the 128 identified from the UPLC-MS analysis passed the Lipinski’s Ro5, while 18 compounds, which revealed more than one violation of the rules, were excluded (Appendix A).

### 3.3. Identification of Overlapping Targets of Secondary Metabolites within SEA and STP Databases

In total, 1040 targets from the STP database and 843 targets arising from the SEA database were obtained. The result of the Venn diagram analysis disclosed the presence of 20.6% (322 genes) of the prevalent overlapping targets being common to the two databases (Figure 2).

A total number of 13,395 targets were identified via the GeneCards predictions based on the findings of the retrieval of the T2DM gene targets from the related databases. Mapping the 322 compound-related targets to the CS secondary metabolites revealed 274 (2%) common targets directly related to T2DM (Figure 3).

### 3.4. PPI Network Analysis

The 274 CS-T2DM overlapped genes arising from the STRING algorithm revealed 274 nodes connected to a network, with 2011 edges. The average node degree was 14.7, while the average local clustering coefficient and the PPI enrichment *p*-value were 0.448 and <1.0 × 10^−16^, respectively. While the edges are characterized by the number of degrees for each target (with the highest number of degrees meaning the best network); however, five targets, *SLC37A*, *HPSE*, *MLNR*, *GABBR1*, and *PTAFR* (circled in Figure 4), had no interaction with any (Figure 4).

#### KEGG Pathway Enrichment Analysis

The results of the KEGG pathway enrichment analysis on the 274 intersecting targets using the STRING database revealed 13 signaling pathways implicated in T2DM associated with the genes related to the CS constituents (Table 1). The identification of the different signaling pathways implicated in T2DM related to the identified genes of the CS phytoconstituents and T2DM target genes were performed through the use of a bubble chart (Figure 5). The bigger the bubble, the lower the false discovery rate of the signaling pathway and the more significant. Thus, the cAMP pathway was the key signaling pathway with the highest significant degree (26), a good strength score (0.95), and the lowest false discovery rate of 1.88 × 10^−14^. A PPI network of the 26 target genes in the cAMP pathway is shown in Figure 6 and there were 54 reported interactions between these 26 nodes, in which the average node degree and average local clustering coefficient are 4.15 and 0.421, respectively. The PPI enrichment *p*-value of the 26 target genes in the cAMP pathway was 2.34 × 10^−11^. However, 4 (*HTR1A*, *PTGER3*, *HCAR2*, *PTGER2*) of the 26 target genes had no interactions with the remaining 22 genes, indicating no connectivity to the network (Figure 6).

### 3.5. Gene Ontology Analysis

The gene ontology analysis performed on the 274 intersecting targets of the secondary metabolites present in the CS and T2DM revealed 494 biological processes, 75 cellular components, and 296 molecular functions. Out of the 494 biological processes reported, the top 10 were identified (Figure 7a), where drug response reported a degree of 30 with the lowest *p*-value of 1.3 × 10^−16^. Additionally, of the 74 cellular components reported, the top 10 components were identified (Figure 7b) and an extracellular exosome was observed with a degree of 78 with the lowest *p*-value of 7.7 × 10^−16^. Similarly, from the 296 molecular functions, ion binding was among the top 10 functions, having a degree of 174 and the lowest *p*-value of 2.8 × 10^−18^ (Figure 7c).

### 3.6. Compound–Target Pathway Network Analysis

Compound–target pathway network analysis revealed 63 nodes (26 related to the cAMP signaling pathway and 37 related to the secondary metabolites present in the CS) interacting with one another through 87 edges (Figure 8a). Additionally, it was revealed that the target *ATP1A1* had no interactions with any of the compounds or metabolites and was excluded from further analysis (Figure 8a). The gene–compound interaction networks revealed that adenosine A1 receptor (*ADORA1*) (Figure 8b), hydroxycarboxylic acid receptor 2 (*HCAR2*) (Figure 8c), and gamma-aminobutyric acid type B receptor subunit 1 (*GABBR1*) (Figure 8d) target genes connected to the highest number of bioactive secondary CS metabolites (15, 11, and 8, respectively).

### 3.7. Molecular Docking Analysis of Identified Secondary Metabolites Present in CS against ADORA1, HCAR2, and GABBR1 in the cAMP Signaling Pathway

The results of the molecular docking analysis for the identified phytoconstituents present in the CS against adenosine A1 receptor (*ADORA1*), hydroxycarboxylic acid receptor 2 (*HCAR2*), and gamma-aminobutyric acid type B receptor subunit 1 (*GABBR1*) targets arising from the NP analysis are presented in Table 2. Quing hau sau, phaseic acid, and tetradecanedioc acid reported the highest negative docking scores against *ADORA1*, *HCAR2*, and *GABBR1*, respectively. All the secondary metabolites present in the CS had higher or equal negative docking scores against *ADORA1*, *HCAR2*, and *GABBR1* compared to metformin (reference standard), except glutaric acid against *HCAR2*. Quing hau sau and phaseic acid as well as caffeic acid were also found to have the highest negative docking scores as compared to resveratrol (reference standard) against *ADORA1* and *HCAR2*, respectively. While the gene agonists 2-Chloro-n6-cyclopentyladenosine, butyric acid, and baclofen were docked against *ADORA1*, *HCAR2*, and *GABBR1*, respectively, quing hau sau and cyperine were observed to reveal most negative docking scores against *ADORA1*. This is in comparison to 2-Chloro-n6-cyclopentyladenosine, with all the compounds exhibiting higher docking scores against *HCAR2* relative to the gene agonist, butyric acid, and an observed reversed trend against *GABBR1* (Table 2).

### 3.8. Molecular Dynamics (MD) Simulation of Identified Secondary Metabolites against ADORA1, HCAR2, and GABBR1 Genes from the cAMP Signaling Pathway

The free binding energies of the top five CS compounds against each of the investigated targets following a 120 ns of MD simulation is presented in Table 3. Against *ADORA1*, all the top five compounds have higher binding free energy compared to the reference standards. Conversely, only 4-hydoxycinnamic acid had lesser free binding energy relative to the standards against *HCAR2*. A partly similar trend was observed in *GABBR1*, with quinic acid and Xi-2,2,6, trimethyl-1,4-cyclohexanedione having lesser binding free energy than the reference standards. In summary, gallicynoic acid B (−48.74 kcal/mol), dodecanedioc acid (−34.53 kcal/mol), and tetradecanedioc acid (−36.80 kcal/mol) had the highest binding affinities for *ADORA1*, *HCAR2*, and *GABBR1*, respectively (Table 3).

The structural and conformational alterations resulting from the binding of the CS phytoconstituents to the elucidated targets were evaluated by the different thermodynamic parameters (Table 4). The average RMSD (4.10 Å) of the apo-gene *ADORA1* was lower compared to the *ADORA1* complexes (standards and compounds) except ginsenoyne E (3.48 Å) and Quing hau sau (4.06 Å) (Table 4). Additionally, after 5 ns of simulation, when the atoms in each system had equilibrated, the fluctuation began thereafter within 2 Å and 7.5 Å till the end of the simulation (Figure 9a). With respect to *HCAR2*, the bound systems of some CS phytocompounds revealed hyped average RMSD values relative to the apo–*HCAR2* (9.64 Å) except phaseic acid–*HCAR2* (7.08 Å), dodecanedioc acid–*HCAR2* (7.80 Å) and 4-hydoxycinnamic acid–*HCAR2* (9.46 Å); in fact, the RMSD values of these three compounds were lower compared to metformin (9.54 Å) and resveratrol (9.10 Å), which was lower compared to the latter compound (Table 4). There was a significant fluctuation in the *HCAR2* system relative to the other genes, with caffeic acid contributing to most of the observed fluctuation between 10 Å and 12.5 Å (Figure 9b). Against *GABBR1*, the unbound system (1.97 Å) was lower compared to the bound systems of the CS phytocompounds and standards, except tetradecanedioc acid (1.54 Å) and quinic acid (1.69 Å) (Table 4). The system fluctuates between 1 Å and 3.5 Å after equilibrating at 16 ns, while the lowest and highest fluctuation in the *GABBR1* system was observed in tetradecanedioc acid and dodecanedioc acid, respectively (Figure 9c).

The top five compounds and resveratrol in complex with *ADORA1* had higher average RMSF values than the unbound *ADORA1* (1.93 Å) and metformin–*ADORA1* (1.88 Å) (Table 4). The compounds under investigation exhibited random fluctuations upon binding to *ADORA1*, with noticeable fluctuations at residues 230–270 between 1.0 Å and 6.5 Å (Figure 10a). The bound systems decreased, with fluctuations between 1 Å and 5.8 Å until amino acid residue 25, before a major increase above 12 Å after residue 350. Contrary to the trend in *ADORA1*, the average RMSF of the apo gene (*HCAR2*) was higher (2.57 Å) compared to the bound systems (CS compounds and resveratrol) except 4-hydoxycinnamic acid–*HCAR2* (2.85 Å) and metformin (2.63 Å) had higher mean RMSF values compared to the apo–*HCAR2* (2.57 Å). Most of the CS compounds revealed reduced RMSF values compared with resveratrol. Minimal fluctuation was observed in the RMSF plot between 1 Å and 5 Å until residue 300 till the end of the simulation, with increased fluctuation between 8 Å to 15 Å (Figure 10b). Similarly, the unbound *GABBR1* (1.29 Å) had lower average RMSF values relative to the bound GABBR1 complexes [dodecanedioc acid–*GABBR1* (1.49 Å), quinic acid–*GABBR1* (1.34 Å), xi-2,2,6, trimethyl-1,4-cyclohexanedione–*GABBR1* (1.58 Å) and metformin–*GABBR1* (1.63 Å)] except methylisocitric acid–*GABBR1* (1.27 Å), tetradecanedioc acid–*GABBR1* (1.25 Å), and *GABBR1*–reservatrol (1.22 Å).There was a reduced swaying between residues 0 and 225, fluctuating in the range of 0.75 Å to 3 Å, while higher fluctuations were noticed around residues 225, 250, 280, and 310 from 0.5 Å to 3.5 Å (Figure 10c).

The mean RoG values of the cyperine–*ADORA1* (29.04 Å) and reservatrol–*ADORA1* (29.16 Å) complexes was higher than the apo-gene (28.64 Å). However, the other bound complexes and metformin revealed RoG values lower than those of *ADORA1* (Table 4). An inconsistency in the stability of all the systems was observed for the initial 30 ns, after this time, the stabilities of the individual systems, particularly metformin, appear to tend towards 120 ns, except ginsenoyne E (Figure 11a). In the same vein, against *HCAR2*, the RoG values of the bound systems of the CS compounds (including dodecanedioc acid, which was marginally lower) and resveratrol were reduced, compared with *HCAR2* (24.24). Caffeic acid (22.98 Å) was the lowest among the co-compounds and resveratrol (23.79 Å). A reduced trend in the stabilities of the systems at around 20 ns was observed, which was then stable throughout the rest of the simulation (Figure 11b). Against GABBR1, the RoG values of all the CS compounds and metformin were higher than the apo-gene (23.11 Å), and resveratrol (22.69 Å) was the lowest (Figure 11c).

The intramolecular hydrogen bonds formed between the complexes were analyzed and are depicted in (Figure 12a–c). Typically, against *ADORA1*, a reduction in the number of hydrogen bonds formed was observed across all the complexes, including cyperine (167.97), domesticoside (162.94), gallicynoic acid B (171.58), ginsenoyne E (170.93), and quing hau sau (170.30), relative to apo–*ADORA1*, with the highest average number of hydrogen bonds (173.53). The lowest number of hydrogen bonds was observed in metformin (136.77) and resveratrol (143.60) (Figure 12a). On the contrary, for *HCAR2*, there was an increase in the number of hydrogen bonds in the bound systems of the CS compounds relative to the unbound *HCAR2* (158.21); this was also observed for the reference standards, except metformin (156.60) (Figure 12b). The increase in the number of hydrogen bonds was expressed in this order: phaseic acid (158.70) < sebaic acid (160.16) < dodecanoic acid (160.98) < 4-hydoxycinnamic acid (163.67) < caffeic acid (166.40) (Table 4). A similar trend was observed with *GABBR1* when the bound systems of *GABBR1* and the CS compounds and standards were higher compared to the apo-gene (203.89) (Figure 12c).

The investigation of the complexes was extended to include an analysis of the solvent accessibility and surface area (SASA). Against *ADORA1*, the mean SASA values of cyperine (22,745.86 Å), domesticoside (22,760.33 Å), and quing hau sau (22,366.55 Å) are marginally higher compared to the unbound *ADORA1* (22,151.21 Å), though gallicynoic acid B-*ADORA1* (21,996.14 Å) and ginsenoyne E-*ADORA1* (21,484.73 Å) had the lowest SASA values among the CS compounds. However, the mean SASA value observed for the reference standards, and the metformin–*ADORA1* (17,546.87 Å) and resveratrol–*ADORA1* (18,462.50 Å) complexes are lesser compared to apo–*ADORA1* (Figure 13a). Furthermore, for *HCAR2*, the CS compound bound complexes reflected reduced SASA values relative to the apo-gene (20,865.81 Å). While the SASA values of the reference standards [metformin (20,894.76 Å), resveratrol (20,736.21 Å)] are marginally lower than the apo-gene, caffeic acid (19,469.68 Å) depicted the lowest SASA values. A downward trend around 15 ns was observed for all the systems, which became stable until the end of the simulation period (Figure 13b). Similarly, the complexes of the CS compounds and standards were higher compared with the apo-gene (17,313.17 Å), except for quinic acid (17,117.23 Å). An inconsistency of all the systems was witnessed throughout the simulation period (Figure 13c).

The data obtained regarding the interaction plots of the investigated CS metabolites (based on the results of the thermodynamics profiles) against each of the established target genes (*ADORA1*, *HCAR2*, and *GABBR1*) revealed diverse bond types such as hydrogen bonds (conventional, carbon, and π-donor), attractive charge, van der Waals, amide π-stacked, π-sigma, π-cation, π-anion, π-alkyl, alky, π-sulphur, salt bridge, unfavorable acceptor–acceptor and donor–donor interactions (Figure 14, Figure 15 and Figure 16; Appendix A). Specifically, the binding of gallicynoic acid B with *ADORA1* after a 120 ns simulation period showed 20 interactions, consisting of 2 hydrogen bonds (PHE168 and HIE346), 2 carbon hydrogen bonds (GLU167 and THR345), 12 van der Waal (ILE60, ASN67, VAL80, ALA81, VAL84, THR88, CYS166, GLU169, TRP315, LEU318, HIE319 and ILE342), 3 alkyl (ALA63, ILE64 and LEU84), and 1 π-anion interaction (TYR339) (Figure 14a). The metformin–*ADORA1* complex had eight interactions (with six amino acid residues), including one hydrogen bond interaction (GLU160), two carbon–hydrogen bonds (GLY160 and GLU161), three van der Waal interactions (LEU 146, TRP153 and PRO162) and two salt bridges with attractive charge interactions (GLU160 and GLU161) (Figure 14b). At the end of the 120 ns simulation period, resveratrol was unbound to *ADORA1*, and thus, displayed no interaction (Figure 14c). The binding of dodecanedioc acid to *HCAR2* following the 120 ns simulation period revealed 20 interactions which consisted of 5 hydrogen bonds (TYR87, 2 SER179, LEU280, and TYR284), 11 van der Waal forces (LEU83, LEU104, ASN110, ARG111, LEU162, SER181, PHE193, ARG251, PHE277, SER281, and THR283), 2 alkyl groups (LEU107 and ALA108) and 2 π-cation interactions with attractive charges (PHE180 and GLU190) (Figure 15a). While, at 60 ns and 120 ns, metformin had no interaction with *HCAR2* as the ligand was unbound (Figure 15b), resveratrol bound to *HCAR2* had eight interactions including five van der Waals (TRP50, PHE54, HIE55, LEU308, and GLY340), two π-alkyl (ALA341 and PRO342) and one π-cation interaction (ARG339) (Figure 15c). Tetradecanedioc acid and *GABBR1* bound presented 13 interactions consisting of 3 hydrogen bonds (ALA126, ARG133, and GLU204), 5 van der Waals (SER106, THR127, THR158, TYR203, and ILE229), 2 alkyl (VAL154 and PHE155), 1 π-sigma (TRP231) and 2 π-anion with salt bridge interactions (HIE129 and ARG133) (Figure 16a). At the end of the 120 ns simulation, metformin was unbound and had no interaction with GABBR1 (Figure 16b). However, resveratrol was bound to *GABBR1* through 12 interactions containing 8 van der Waals (TRP18, SER106, HIE123, HIE129, GLN150, GLN152, THR158, and TRP231), 2 π-alkyl (ALA126 and VAL154) and 2 π-π-stacked interactions (PHE155 and TYR203) (Figure 16c).

### 3.9. Molecular Orbital Properties

A detailed analysis of the top-scoring compounds’ structural and chemical reactivity properties was generated using DFT (Table 5). Apart from resveratrol (4.06 eV), which recorded the lowest energy gap across the three targets, and metformin (5.26 eV), which was second to the lowest against *GABBR1*, ginsenoyne E (4.43 eV), caffeic acid (4.14 eV), and quinic acid (5.51 eV) exhibited the lowest energy gap against *ADORA1*, *HCAR2*, and *GABBR1*, respectively (Figure 17). Consequently, ginsenoyne E (0.45 eV), caffeic acid (0.48 eV), and quinic acid (0.36 eV) exhibited the highest chemical softness in a similar manner, with the energy gap from xi-2,2,6-trimethyl-1,4-cyclohexanedione also having the lowest chemical softness against *GABBR1*, the same as quinic acid. However, the highest chemical hardness value was observed in quing hau sau (3.03 eV), sebaic acid (3.70 eV), and tetradecanedioc acid (3.70 eV). Furthermore, the highest electronegativity value was observed in ginsenoyne E (4.80 eV), phaseic acid (4.50 eV), and methylisocitric acid (4.21 eV) against *ADORA1*, *HCAR2*, and *GABBR1*, respectively, while dodecanedioc acid (1.38 eV) exhibited the lowest electrophilicity value against *HCAR2* and *GABBR1*, with cyperine (1.47 eV) showing the lowest against *ADORA1*.

## 4. Discussion

Historically, medicinal plants, for many centuries, have continued to be employed in traditional medicine systems for the provision of knowledge and relieve and/or cure numerous diseases and illnesses [4], most especially T2DM [7,10]. Corn silk, for example, despite being an abundant waste material, is a potential remedy for T2DM [8,10] and established in many studies, including antioxidant and anti-inflammatory activities [11,16,17,18,48].

While the therapeutic significance of medicinal plants (such as CS) is due to be endowed with key phytochemicals, if the development of novel drugs [49] is to be warranted, it is crucial, therefore, to begin the screening of plants of potential therapeutic significance to determine their metabolite profiles [41]. However, it must be noted that the type of solvents utilized in the extraction of medicinal plants and/or natural products is critical in determining the type of the composition and concentration of phytoconstituents [50]. The choice of polar solvents in the study, though majorly used in indigenous medicine for the extraction and preparation of formulations, is based on established literature reports of a possible high extraction yield [50,51]. However, there are reports of moderate apolar (ethyl acetate) and non-polar solvents (acetone and hexane) in addition to the polar solvents used on CS [50,51,52,53,54].

The high abundance of (7′R)-(+)-Lyoniresinol 9′-glucoside, cnicin, methyisocitric acid, chrysoeriol 4′,7-diglucoronide, and 3-isopropylmalatte in the aqueous extract and D-leucic acid, dodecanedioc acid, quercetin-3-(2″,3″,4″-triacetylgalactoside), sebacic acid, and p-coumaroyl malic acid in the ethanolic extract and vice versa could be attributed to the degree of polarity of the extracting medium [55] and may explain or buttress the variation (46.8%) in the types and amounts of phytoconstituents (generally) as identified based on UPLC-MS analysis. While the use of this technique is adequate and reliable, as buttressed in the work of Fougre et al. [56], adopting a related tool, the high abundance of phytoconstituents, namely, azealic acid, isowertin 2″-rhamnoside, D-2-hydrozyglutaric acid, citraconic acid, 3-p-courmaroylquinic acid, cis-aconitic acid, UNPD129404, caffeic acid ethyl ester, and myricitrin, particularly in the raw CS sample, may be suggested to contribute to the pharmacological attributes (glucose lowering) of CS, since the concentration of bioactive ingredients within a medicinal sample is well-established to influence the pharmacological effectiveness. In fact, compounds or their derivatives such as azealic acid, 3-p-coumaroylquinic acid, caffeic acid, and myricitrin identified in CS have also been detected in several other plants exhibiting an antihyperglycaemic effect [57,58,59,60].

The network pharmacology approach provides avenues for new drug candidates or secondary metabolites, genetic target profiles, and connected signaling pathways linked to diseases, including infectious and non-infectious ones [30,61], achieved by a number of analyses [25,27,28,30,31]. While the five targets not connected in the PPI network may indicate their lack of involvement in the connectivity of the network, suggesting that they may not necessarily or unlikely be involved in any metabolic pathway or offer any molecular function, the cAMP as a key second messenger in signaling pathways has been reported, particularly in drug development [62], particularly against T2DM [25]. cAMP maintains glucose homeostasis in many ways, including insulin and glucagon secretion, glucose uptake, glycogen synthesis and breakdown, gluconeogenesis, the maintenance of β-cell differentiation, and the neural control of glucose homeostasis [63]. The establishment of cAMP as the best signaling pathway in this study buttresses it as an important signaling pathway to consider if deciphering the MoA of the CS in alleviating the negative effect of hyperglycaemia is to be completely elucidated [64].

The gene ontology analysis of the 274 intersecting targets between the CS phytoconstituents and T2DM targets revealed the regulation of the drug response, exosomes, and ion binding as the most significant biological process, cellular component, and molecular function. The regulation of the drug response modulates the frequency, rate, or extent of the drug response and refers to the regulation of drug resistance and determines the response and side effects a drug has on the body [65,66,67,68]. Exosomes are delivery vehicles for different signaling molecules (lipids, proteins, and nucleic acids) and serve as important mediators of intracellular communication [66]. Extracellular exosomes play a role in the regulation of inflammation, the stimulation of glycogen accumulation, and the regulation of GLUT4 metabolism, all of which are implicated in T2DM [69,70]. Ions are involved in the folding of proteins and nucleic acids, enzyme catalysis, and numerous cellular signaling processes, and thus, ion binding has a significant role in the normal functioning of processes in the human body [65,71]. A change in the binding of ions in the human body can affect many processes involved in glucose metabolism, including insulin signaling, secretion, and β-cell functioning [71]. Several studies have previously explored the role that the regulation of drug response, exosomes, and ion binding plays in the pathogenesis of T2DM [66,69,70,72,73].

Compound–target pathway network analysis is important for the discovery of therapeutic targets as well as lead compounds [74]. The association of many of the CS compounds to cAMP targets such as *ADORA1*, *HCAR2*, and *GABBR1* highlighted them among the genes in the cAMP pathway. Adenosine A1 receptor (*ADORA1*), a G protein-coupled receptor, inhibits the enzyme adenylate cyclase and plays a role in the regulation of cell metabolism and gene transcription, and therefore, has been identified as an important drug target for the treatment of various diseases and illnesses [75,76], including T2DM via glucose homeostasis and glucagon secretion regulation [77]. The identification of *ADORA1* as a key target for T2DM and related complications (e.g., nephropathy) therapy buttresses previous studies (on morusin, kuwanon C, and morusyunnansin) from *Morus alba* (leaves) and *Salvia miltiorhiza* through NP and molecular docking analyses [77,78]. Hydroxycarboxylic acid receptor 2 (*HCAR2*) is a G-protein-coupled receptor responsible for mediating the antilipolytic actions of niacin and the lowering of blood lipid levels [79]. The expression of *HCAR2* in cells affects the regulation of inflammatory factors, inhibition of lipolysis, and glucose homeostasis [79,80]. G-protein-coupled receptors (GPCRs), including *GABBR1* are transmembrane signaling molecules involved in a wide variety of physiological processes such as the modulation of insulin secretion and the regulation of islet function, making them potential targets for antidiabetic compounds [81]. Previous reports of their involvement as a therapeutic target for treating various diseases such as T1DM, cancer, etc., have been well-established [81,82,83,84,85]. Typically, studies have demonstrated that, in human beta cells, signaling through *GABBR* participates in an autocrine feedback inhibition loop that regulates beta cell-specific gene expression and insulin secretion [84]. Although *GABBR1* was among the five non-target genes of the PPI network, the compound–target pathway network analysis revealed its importance as a therapeutic target, having the third highest number of CS phytoconstituents related to it [25,29].

Molecular docking, a structure-based drug design approach, is a preliminary screening tool for the identification of a suitable ligand (say, secondary metabolites) based on its binding free energy (docking score) at the active site of the target gene that could be developed as a probable candidate [86]. The most negative docking score of Quing hau sau, phaseic acid, and tetradecanoic acid, arising from their binding at the active sites of *ADORA1*, *HCAR2*, and *GABBR1*, respectively, suggests their binding affinities and superiority (particularly the former two) compared to the reference standards. This is because the higher the negative binding free energy, the better the fitness of that bioactive compound [86,87], and thus suggests an attraction between the CS secondary metabolites and the target genes [30]. While no related report of the possible stability of phytocompounds with the studied targets (particularly) *ADORA1* and/or superiority over reference standards against the target of concern, the overall findings of the molecular docking analysis provide an insight into the further investigation of CS bioactive metabolites as possible lead antidiabetic compounds.

Since molecular docking only predicts a metabolite’s fitness at a protein active site [35], which limits its consideration as a measure of stability, binding free energy value calculations and MD simulation were employed to evaluate the compound-to-protein target systems and binding conformation data [45]. The highest negative ∆G_bind_ reported for gallicynoic acid B, dodecanedioc acid, and tetradecanedioc acid is indicative of significant binding affinity to *ADORA1*, *HCAR2*, and *GABBR1*, respectively, and could represent greater interactions with the targets compared to other investigated metabolites and standards. The observed lower binding free energy of some top-ranked metabolites against *ADORA1*, *HCAR2*, and *GABBR1* compared to metformin and resveratrol indicates their better potential to inhibit the respective targets relative to the standards used.

The binding of ligands (say CS phytoconstituents) may constitute structural or conformational changes in the respective targets, which could ultimately alter their biological activities [45,61]. The post-dynamics simulation evaluates the likely conformational changes of the protein as a result of the binding of the ligand, which are mostly determined by parameters such as RMSD (stability), RMSF (flexibility), RoG (compactness), SASA (degree of hydrophobic interactions), and intermolecular H bonding [88,89,90]. The revealed post-MDS analysis of the bound and unbound complex systems for the CS metabolites against *ADORA1*, *HCAR2*, and *GABBR1* targets in terms of the average RMSD (value) expresses the degree of convergence, stability, or deviations produced by a protein in a simulation system [91]. The lowest average RMSD values observed in ginsenoyne E–*ADORA1*, dodecanedioc acid–*HCAR2* and tetradecanedioc acid–*GABBR1* complexes are suggestive of the ability of the phytocompounds to enhance the stability of the target. Although the increased mean RMSD values observed in this study for most of the compounds including the reference standards, particularly against *ADORA1* and *HCAR2*, were above the satisfactory or acceptable 3.0 Å [89,92]. However, the average RMSD values of several phytoconstituents lower than or comparable to the respective standards may indicate that the abovementioned CS phytoconstituents may have better potential in promoting the structural stability of the genes [93,94], though an RMSD value above may suggest an unstable complex and the inability of the compounds to inhibit the target protein (and, in this case, *ADORA1* and *HCAR2*) [88]. Meanwhile, the mean RMSD values of CS phytoconstituent–GABBR1 were less than 3.0 Å, suggestive of the good structural stability and compatibility of the metabolites with the gene [30,43,90,95].

The average RMSF (value) indicates the effect the bound compound has on active site residue behavior, with lower or higher alpha (α)-carbon (C) shifts indicating less or more flexible movements, respectively [94,96,97]. A lower RMSF value indicates that the created intra- and intermolecular bonds are more stable [97]. The reduced RMSF of quing hau sau against *ADORA1* compared to other phytocompounds with increased RMSF relative to resveratrol indicates lesser flexible movements, and thus, the greater stability of the complex [29]. Except for 4-hydroxycinnamic acid, the lesser average RMSF values observed in all the investigated phytocompounds against *HCAR2* compared to the standard *HCAR2* complexes indicated less flexible movements. This observation suggests the lesser flexibility of *HCAR2* amino acid residues, following the binding of the top-ranked metabolites, revealing their stronger attraction and ability to promote *HCAR2* amino acid residue stability [97,98,99]. Similarly, the reduced RMSF of methylisocitric acid and tetradecanedioc acid bound to *GABBR1* compared to metformin suggests less flexibility and stability (of the complexes) [35,44,93,95].

The RoG (value) evaluates the overall structural compactness of molecule–target complex systems, which may affect the biological properties due to induced changes from the ligand binding to a target [46,94,100,101]. The lowest or comparable average RoG values of the CS compounds (particularly, gallicynoic acid B, caffeic acid, and methylisocitric acid) against *ADORA1* and *HCAR2* relative to the standards, respectively, is indicative of a greater stability [30]. Typically, the lower RMSD of a ligand (such as CS compounds excluding cyperine) compared to resveratrol against *ADORA1* is suggestive of a superior stability compared to the latter.

During a simulation, the number of hydrogen bonds in a protein may be calculated, thus, providing insight into how ligand binding affects the protein’s stability [90,94,102,103]. The observed reduction (for this study) in the number of H bonds in the CS compounds (most importantly, domesticoside) and reference standards against *ADORA1* following complex formation may be due to an intramolecular breakage of these bonds [94]. This observation is coherent with S’thebe et al.’s [97] report, in which ligand binding to the investigated protein resulted in the reduction in the intramolecular hydrogen bonding. However, the increased number of hydrogen bonds of the bound systems of the CS compounds (especially caffeic acid, tetradecanedioc acid, and resveratrol) against *HCAR2* and *GABBR1* (respectively), as observed in this study, may suggest that the ligands occupy part of the proteins’ intramolecular space. This finding is in tandem with Aribisala and Sabiu’s [45] report, in which ligand binding resulted in an increased number of hydrogen bonds [45].

Protein folding and variations in the surface area are analyzed using the solvent-accessible surface area (SASA), with higher values indicating more surface area and lower values suggesting less protein volume as the simulation progresses [30,44,104]. The lower average SASA values of gallicynoic acid B and ginsenoyne E following binding to *ADORA1* relative to the unbound system indicates the compactness and stability of the complexes, as it shows that more residues in the unbound state are exposed to the solvent [104,105]. The lowest SASA values as observed with the reference standards depicted the inferiority of the CS compounds in terms of their compactness and stability. While the superior compactness and stability of the CS compounds (especially caffeic and phaseic acids) were also confirmed against *HCAR2*, interestingly, the reference standards were inferior to these phytocompounds in terms of stability and compactness, owing to the high SASA values displaced. The reports of phytocompounds exhibiting better stability above the studied standard are well-established [30,61,98]. The comparable SASA values of quinic acid and resveratrol against *GABBR1*, which were the lowest relative to the unbound system and other bound systems, reflects the compactness and stability of the protein–ligand complex [98]. While it is established that ligand–protein interactions may have a major effect on the SASA value [92], the SASA results may be seen to correspond with the findings from other studies, where the binding of the ligands enhanced the thermodynamic stability of the drug targets [29,30]. Above all, the findings (SASA) may be observed to be consistent with the revelations from other thermodynamic parameters (RMSD, ROG, and RMSF).

Various ligand–protein interactions were observed over the 120 ns MD simulation. The hydrogen bonds formed between a ligand and the protein target receptor determine the protein–ligand complex’s stability [45,94,99]. Additionally, hydrogen bonds are indicative of a drug’s specificity, metabolism, and adsorption [89]. Notwithstanding the afore-mentioned, other significant interactions like van der Waals and π-alkyl also contribute to the stability of the investigated complexes [29]. Although van der Waals interactions are weak in comparison to hydrogen bonds, when combined, the strength of the interaction increases significantly [104]. Interestingly, these major bonds form the backbone, contributing to the stability witnessed between CS compounds such as gallicynoic acid B, dodecanedioc acid, and tetradecanedioc acid and their respective target genes (*ADORA1*, *HCAR2*, and *GABBR1*) as compared to the reference standards. In fact, the lack of interaction between resveratrol and *ADORA1* and with *GABBR1* alludes to the structural instability of the complexes, which contributed to the observed low binding free energy of the respective complexes. Since it is a known fact the stability of a drug and/or ligand–protein complex contributes positively to the eventual pharmacological effect of the drug [105], the contribution of these bonds or interactions to the stability of gallicynoic acid B–, dodecanedioc acid–, and tetradecanedioc acid–target complexes highlights the possible superior therapeutic advantage of these CS compounds over the already available standard drug.

The lead compounds were characterized using quantitative chemical parameters to probe into their potential molecular properties of therapeutic importance. Frontier molecular orbitals, namely, the LUMO and HOMO, are critical for identifying the chemical reaction of a system [106]. The energy gap between a molecule’s HOMO and LUMO influences its chemical reactivity, kinetic stability, optical polarizability, and chemical hardness–softness [99,107]. In particular, the reactivity of a molecule exhibits a direct correlation with its energy gap, indicating that a molecule with smaller dimensions possesses a higher propensity to react with other molecular entities, such as proteins and enzymes [107]. The lower energy gap observed in ginsenoyne E, caffeic acid, and quinic acid against *ADORA1*, *HCAR2*, and *GABBR1*, respectively, suggests their high reactivities relative to other CS metabolites’ reactivity (Table 4). Furthermore, molecules characterized by a larger energy gap tend to display enhanced hardness, reducing their reactivity [99,108]. The chemical hardness serves as a robust metric for evaluating the chemical stability of a molecule and plays a crucial role in investigations pertaining to drug design elucidations [36,99]. Soft molecules are characterized by an elevated level of polarizability compared to hard molecules, primarily because of their reduced energy demand for excitation [108]. Expectedly, the high reactivity of ginsenoyne E, caffeic acid, and quinic acid against *ADORA1*, *HCAR2*, and *GABBR1*, respectively, could equally be linked to their high chemical softness value. On the contrary, the highest chemical hardness value exhibited by quing hau sau, dodecanedioc acid, and tetradecanedioc acid against *ADORA1*, *HCAR2*, and *GABBR1*, respectively, suggest their resistance to charge transfer and are the least reactive, as evidenced by their relatively high energy gap (Appendix A). In addition, the observed reduced value of the chemical potential across all the compounds implies that the molecules exhibit diminished polarization. As a result, this molecule displays heightened resilience against electronic deformation when exposed to minor perturbations during a chemical reaction. A negative chemical potential value further supports the molecule’s stability by inhibiting spontaneous decomposition [107,109]. Another cDFT descriptor is electronegativity, a fundamental metric that quantifies electron distribution within a given molecular entity [105]. The top-scoring compounds’ readiness to accept electrons is revealed in their high electronegativity values against the investigated targets [109]. Moreover, the electrophilicity index quantifies a molecule’s propensity to accept electrons from the surrounding environment, thus indicating its inherent capability to act as an electrophile [108]. A molecule may be categorized as a low electrophile if its electrophilicity value falls below 0.8 eV, while a moderate electrophile is characterized by an electrophilicity value ranging from 0.8 to 1.5 eV, and a molecule is deemed a heavy electrophile when its electrophilicity value exceeds 1.5 eV [36,109]. Interestingly, the electrophilicity index of the top-scoring compounds suggests a significant electrophile presence around the molecules except in cyperine, and dodecanedioc acid with a moderate electrophile presence.

## 5. Conclusions

The metabolomic profiling identified 128 secondary metabolites in various samples of CS (raw and three extracts) with qualitative and quantitative variance in the type and amounts of secondary metabolites. This is attributed to the polarity of the solvents, namely, water, hydro-ethanol, and ethanol, used for the extractions. The network pharmacology analysis identified 274 common overlapping target genes related to the CS phytoconstituents and T2DM. The top-scoring metabolites identified in the various extracts of CS are majorly involved in the modulation of the cAMP signaling pathway, which is implicated in glucose metabolism and homeostasis. The therapeutic target genes *ADORA1*, *HCAR2*, and *GABBR1* in the cAMP pathway were related to most of the CS phytoconstituents, with *ADORA1* being related to more than the others. This suggests the CS phytoconstituents modulate the activity of *ADORA1*, *HCAR2*, and *GABBR1* in the cAMP pathway. The molecular docking analysis identified several CS phytoconstituents as inhibitors of the target genes (*ADORA1*, *HCAR2*, and *GABBR1*) from the cAMP pathway. Based on the structural stability and affinity of several CS phytoconstituent–target gene complexes: gallicynoic acid B–*ADORA1*, dodecanedioc acid–*HCAR2* and tetradecanedioc acid–-*GABBR1*, these abovementioned phytoconstituents have been identified as the key components of CS that behave as agonists of the cAMP signaling pathway. It is deduced that through the modulation of the therapeutic target genes *ADORA1*, *HCAR2*, and *GABBR1*, present in the cAMP signaling pathway, phytoconstituents present in CS could contribute to the maintenance of glucose homeostasis, the proper functioning of pancreas and pancreatic beta-cells, as well as the prevention of T2DM-associated secondary complications. Therefore, this study contributes to the use of CS as a therapeutic agent for the management of T2DM. Further in vitro and in vivo studies are recommended to validate the findings, and, interestingly, efforts are ongoing in this direction.

## Figures and Tables

**Figure 1 biology-12-01509-f001:**
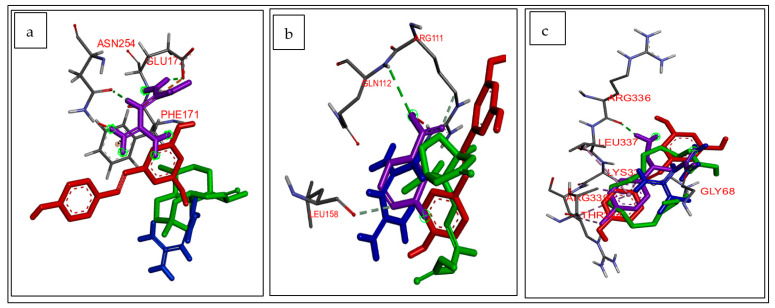
Superimposed structure of (**a**) *ADORA1* (grey) with native inhibitor (purple), metformin (blue), resveratrol (red), and ligand with highest docking score (green); (**b**) *HCAR2* (grey) with native inhibitor (purple), metformin (blue), resveratrol (red), and ligand with highest docking score (green); (**c**) *GABBR1* (grey) with native inhibitor (purple), metformin (blue), resveratrol (red), and ligand with highest docking score (green).

**Figure 2 biology-12-01509-f002:**
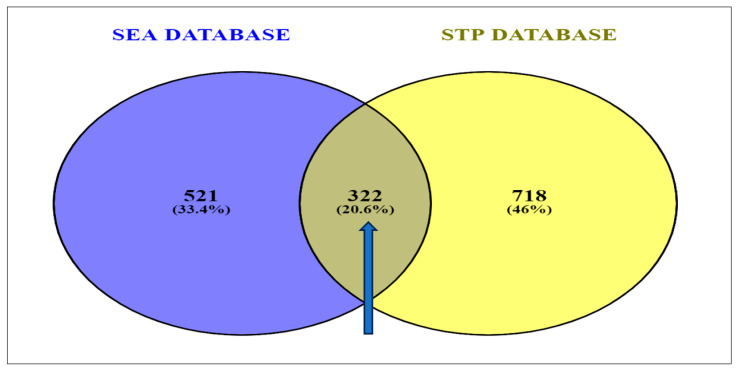
Identification of overlapping targets linked to secondary metabolites present in corn silk between SEA and STP databases [SEA: similarity ensemble approach; STP: Swiss target prediction].

**Figure 3 biology-12-01509-f003:**
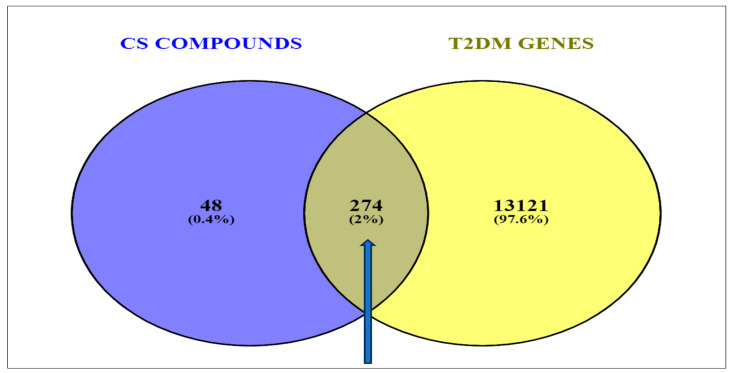
Venn diagram showing the overlapping genes common to the corn silk phytoconstituents and the T2DM-related target genes.

**Figure 4 biology-12-01509-f004:**
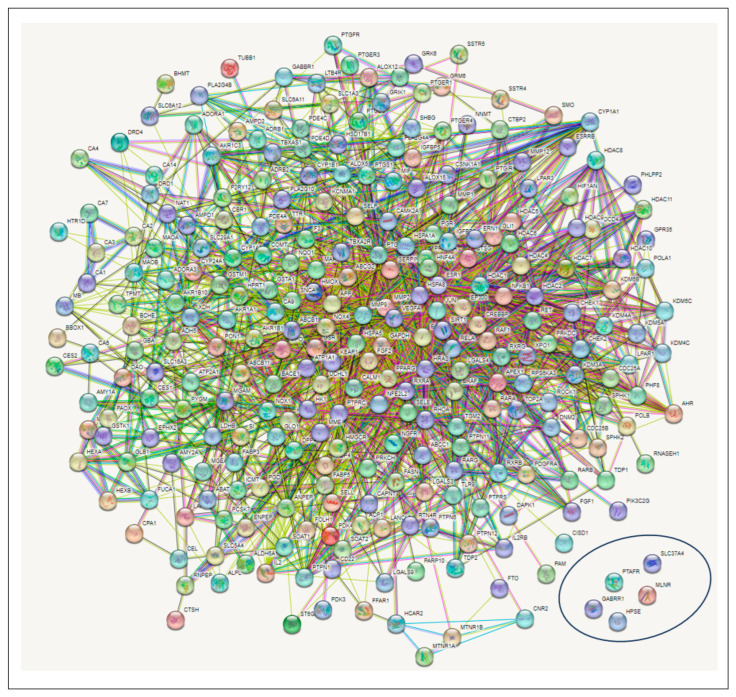
Protein–protein interaction (PPI) network showing the intersecting targets between the secondary metabolites in corn silk and type 2 diabetes mellitus. (Circled target genes showed no interactions with any nodes).

**Figure 5 biology-12-01509-f005:**
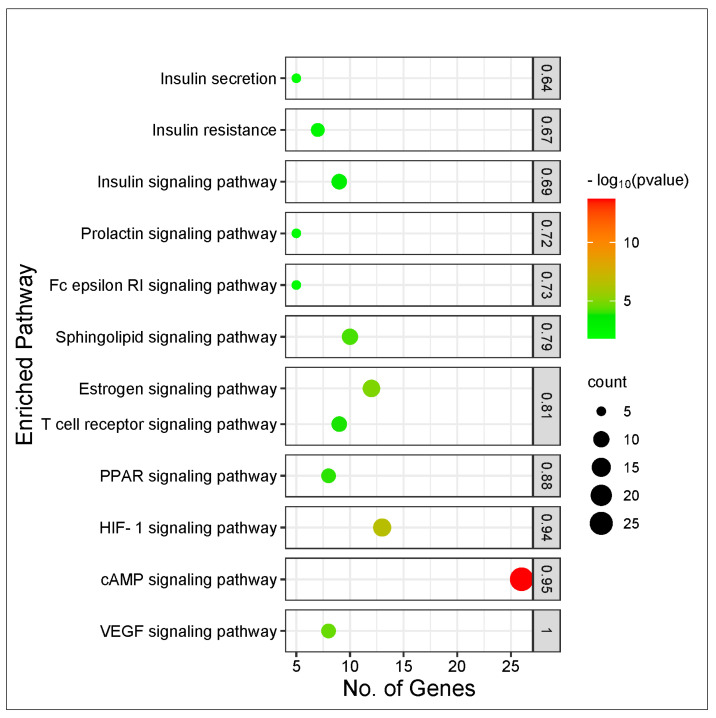
Bubble plot of KEGG pathway enrichment analysis of 274 intersecting targets related to CS secondary metabolites and T2DM.

**Figure 6 biology-12-01509-f006:**
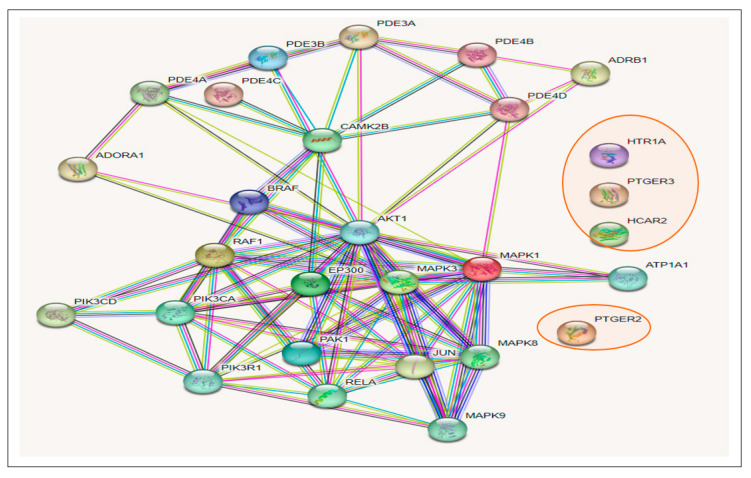
Gene–gene interactions amongst 26 T2DM target genes in the cAMP pathway related to CS secondary metabolites (The 4 target genes not connected to the 22 other genes are circled in red).

**Figure 7 biology-12-01509-f007:**
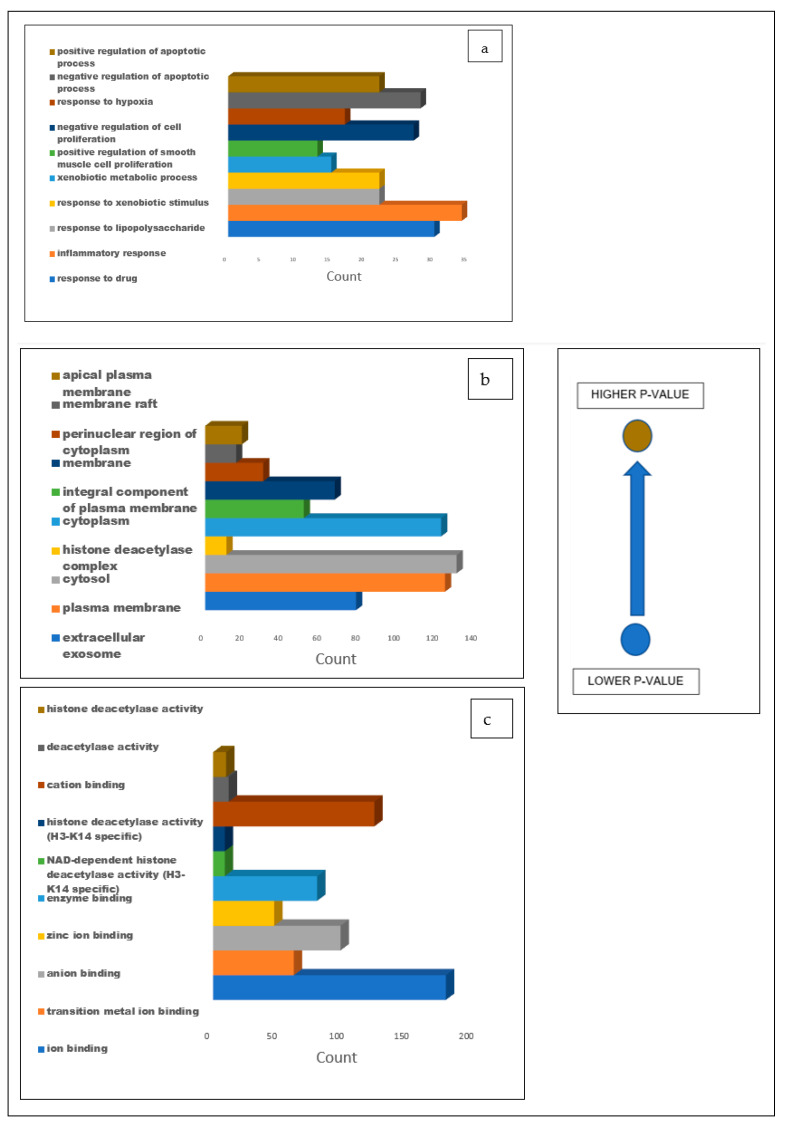
Bar graphs illustrating the GO analysis of the 274 common targets between T2DM and CS secondary metabolites for (**a**) biological processes, (**b**) cellular components, and (**c**) molecular functions with blue representing the lowest *p*-value and light brown representing the higher *p*-value.

**Figure 8 biology-12-01509-f008:**
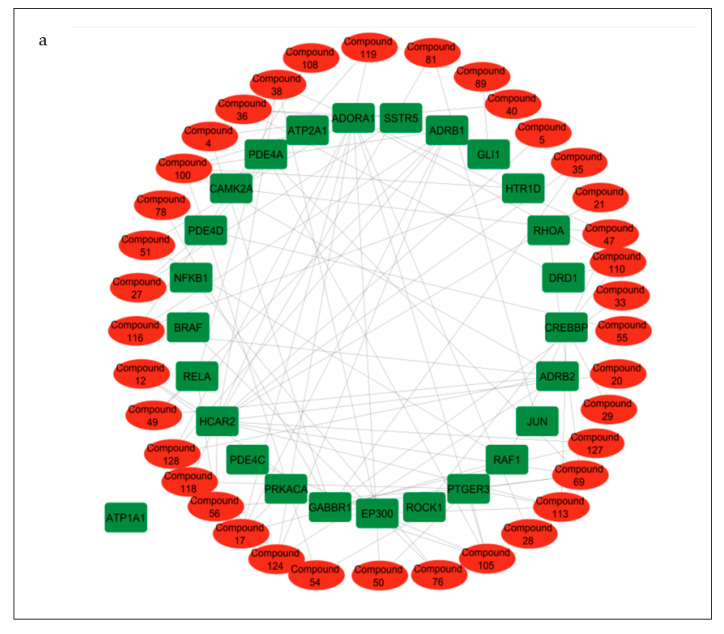
(**a**) Pathway of compound–target interaction network illustrating 37 CS secondary metabolites (red) interacting with 26 T2DM target genes implicated in the cAMP pathway (green); CS secondary metabolites (yellow) interacting with cAMP pathway genes (**b**) *ADORA1* (yellow rectangle); (**c**) *HCAR2* (yellow rectangle); (**d**) *GABBR1* (yellow rectangle).

**Figure 9 biology-12-01509-f009:**
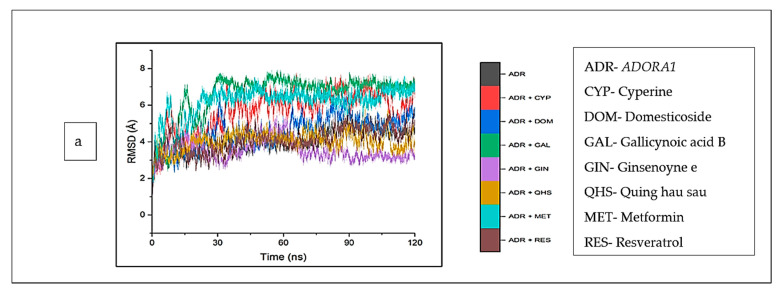
Root mean square deviations plots of comparison between CS phytocompounds, standards and target genes (**a**) *ADORA1*, (**b**) *HCAR2*, and (**c**) *GABBR1* determined over 120 ns simulation.

**Figure 10 biology-12-01509-f010:**
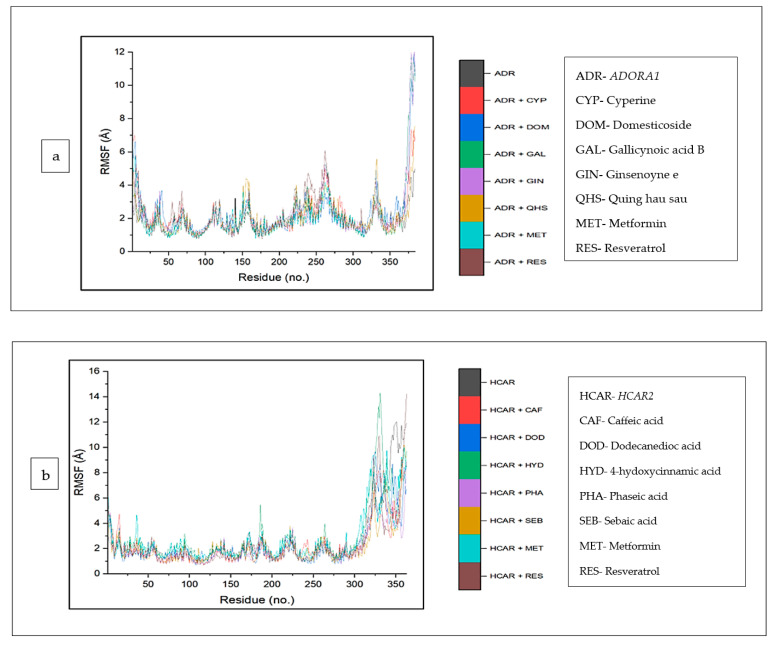
Root mean square fluctuation plots of comparison between CS phytocompounds, standards and target genes (**a**) *ADORA1*, (**b**) *HCAR2*, and (**c**) *GABBR1*, determined over 120 ns simulation.

**Figure 11 biology-12-01509-f011:**
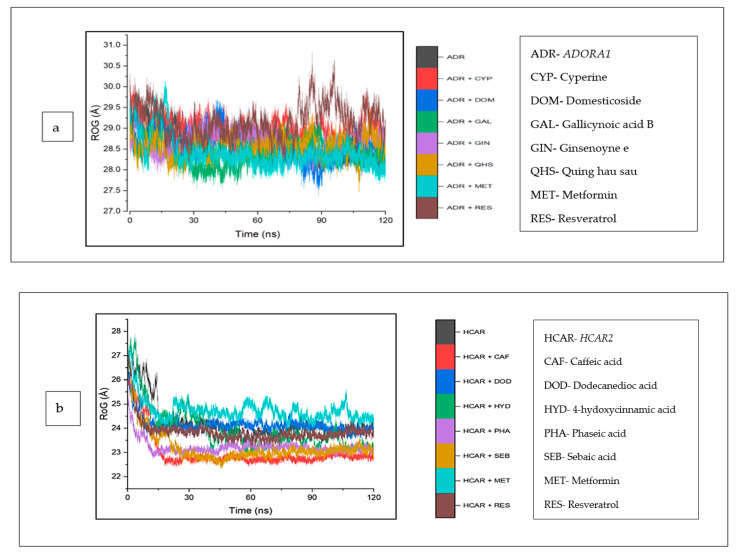
Radius of gyration plots of comparison between CS phytocompounds, standards and target genes (**a**) *ADORA1*, (**b**) *HCAR2*, and (**c**) *GABBR1*, determined over 120 ns simulation.

**Figure 12 biology-12-01509-f012:**
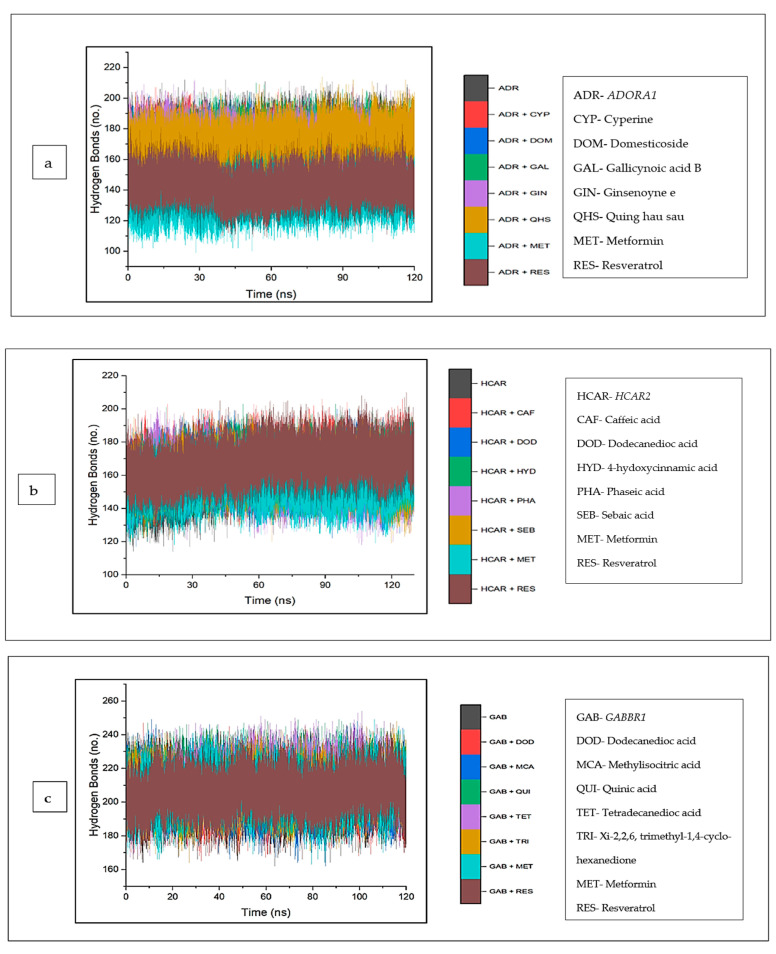
Backbone hydrogen bonding trajectory between CS phytocompounds, standards, and target genes (**a**) *ADORA1*, (**b**) *HCAR2*, and (**c**) *GABBR1* determined over 120 ns simulation.

**Figure 13 biology-12-01509-f013:**
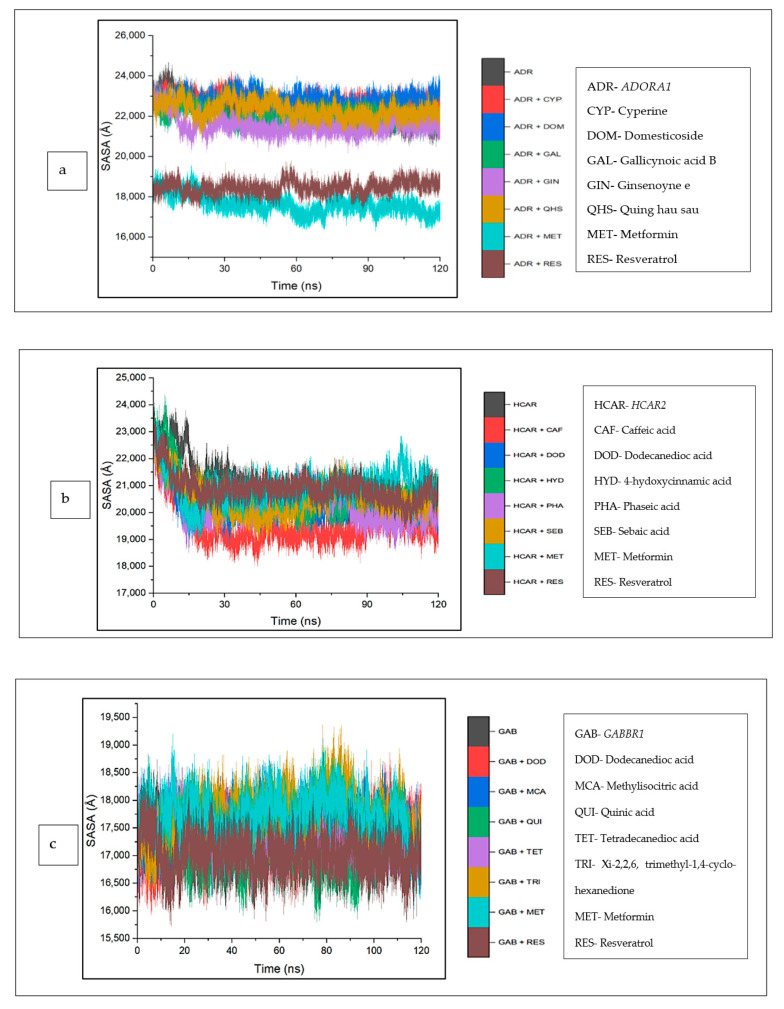
Solvent accessible surface area plots of comparison between CS phytocompounds, standards, and target genes (**a**) *ADORA1*, (**b**) *HCAR2*, and (**c**) *GABBR1*, determined over 120 ns simulation.

**Figure 14 biology-12-01509-f014:**
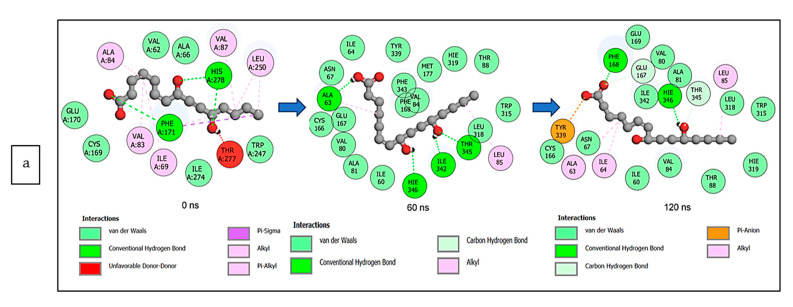
2-D interaction plots of *ADORA1* with (**a**) gallicynoic acid B, (**b**) metformin, and (**c**) resveratrol at 0, 60 and 120 ns.

**Figure 15 biology-12-01509-f015:**
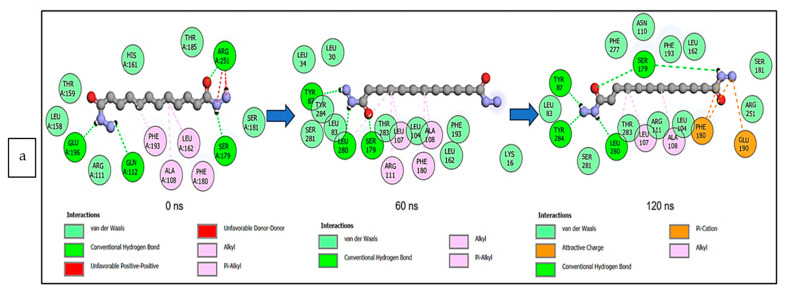
2-D interaction plots of target gene *HCAR2* with (**a**) dodecanedioc acid and standards, (**b**) metformin, and (**c**) resveratrol at 0, 60, and 120 ns.

**Figure 16 biology-12-01509-f016:**
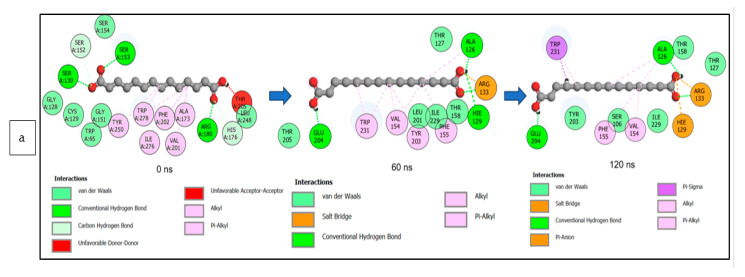
2-D interaction plots of *GABBR1* with (**a**) tetradecanedioc acid, (**b**) metformin, and (**c**) resveratrol at 0, 60, and 120 ns.

**Figure 17 biology-12-01509-f017:**
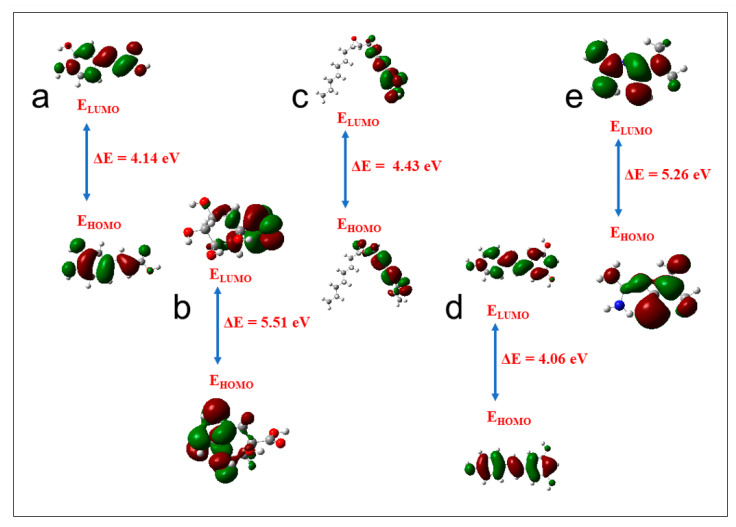
Frontier molecular orbitals for (**a**) caffeic acid, (**b**) quinic acid, (**c**) ginsenoyne, (**d**) reservatrol, (**e**) metformin (E_LUMO_: energy of the lowest unoccupied molecular orbital; E_HOMO_: energy of the highest occupied molecular orbital).

**Table 1 biology-12-01509-t001:** Pathway enrichment analysis results of the 274 intersecting targets involved in 13 signaling pathways implicated in T2DM.

Pathway Code	Description	Degree	Total	Strength	False Discovery Rate	Genes
hsa04933	AGE-RAGE signaling pathway in diabetic complications	11	98	0.91	4.88 × 10^−6^	*MMP2*, *SERPINE1*, *NOX4*, *F3*, *JUN*, *NOX1*, *RELA*, *HRAS*, *VEGFA*
hsa04917	Prolactin signaling pathway	5	69	0.72	8.80 × 10^−3^	*NFKB1*, *RAF1*, *RELA*, *ESR1*, *HRAS*
hsa04915	Estrogen signaling pathway	12	133	0.81	1.08 × 10^−5^	*MMP2*, *RAF1*, *RARA*, *EGFR*, *PRKACA*, *PGR*, *JUN*, *MMP9*, *GABBR1*, *ESR1*, *HRAS*, *HSPA8*
hsa04664	Fc epsilon RI signaling pathway	5	66	0.73	7.50 × 10^−3^	*RAF1*, *PLA2G4A*, *ALOX5*, *PLA2G4B*, *HRAS*
hsa04660	T cell receptor signaling pathway	9	101	0.81	1.30 × 10^−4^	*NFKB1*, *IL2*, *RAF1*, *JUN*, *RELA*, *PTPN6*, *RHOA*, *HRAS*, *PTPRC*
hsa04370	VEGF signaling pathway	8	57	1.00	2.77 × 10^−5^	*SPHK2*, *RAF1*, *SPHK1*, *PLA2G4A*, *PTGS2*, *PLA2G4B*, *HRAS*, *VEGFA*
hsa04071	Sphingolipid signaling pathway	10	116	0.79	6.82 × 10^−5^	*NFKB1*, *SPHK2*, *RAF1*, *SPHK1*, *ADORA1*, *ABCC1*, *ROCK1*, *RELA*, *RHOA*, *HRAS*
hsa04910	Insulin signaling pathway	9	133	0.69	7.50 × 10^−4^	*PYGM*, *RAF1*, *BRAF*, *HK2*, *FASN*, *PRKACA*, *PTPN1*, *HK1*, *HRAS*
hsa04911	Insulin secretion	5	82	0.64	1.59 × 10^−2^	*FFAR1*, *KCNMA1*, *PRKACA*, *CAMK2A*, *ATP1A1*
hsa04931	Insulin resistance	7	107	0.67	3.10 × 10^−3^	*PYGM*, *NFKB1*, *PTPN11*, *MGEA5*, *PTPN1*, *RPS6KA3*, *RELA*
hsa04024	cAMP signaling pathway	26	208	0.95	1.88 × 10^−14^	*NFKB1*, *GLI1*, *RAF1*, *CREBBP*, *EP300*, *BRAF*, *SSTR5*, *ADRB2*, *PRKACA*, *PDE4D*, *PDE4C*, *PTGER3*, *ATP2A1*, *ADORA1*, *ADRB1*, *JUN*, *HTR1D*, *GABBR1*, *PDE4A*, *HCAR2*, *DRD1*, *CAMK2A*, *ROCK1*, *RELA*, *RHOA*, *ATP1A1*
hsa03320	PPAR signaling pathway	8	75	0.88	1.11 × 10^−4^	*FABP4*, *PPARG*, *FABP5*, *MMP1*, *RXRG*, *FABP3*, *RXRB*, *RXRA*
hsa04066	HIF-1 signaling pathway	13	106	0.94	2.63 × 10^−7^	*HMOX1*, *SERPINE1*, *NFKB1*, *GAPDH*, *CREBBP*, *EP300*, *EGFR*, *HK2*, *LDHB*, *CAMK2A*, *RELA*, *HK1*, *VEGFA*

**Table 2 biology-12-01509-t002:** Molecular docking scores of identified secondary metabolites against cAMP pathway genes (*ADORA1*, *HCAR2*, and *GABBR1*).

Target	Compounds	Docking Score (kcal/mol)
*ADORA1*	Quing hau sau	−8.5
	Cyperine	−7.9
	Domesticoside	−6.9
	Gallicynoic acid B	−6.4
	Ginsenoyne e	−6.3
	Caffeic acid	−6.3
	Caffeoyl tartaric acid	−6.3
	Methyl geranate	−6.1
	Tetradecanedioic acid	−5.7
	Traumatic acid	−5.6
	7-acetoxy-5,6-dimethoxycoumarin	−5.6
	Methylisocitric acid	−4.9
	(-)-6-((2S,3R,4R,5S,6R)-3,4-dihydroxy-6-(hydroxymethyl)-5-methoxytetrahydro-2H-pyran-2-yloxy)-8-hydroxy-3-methyl-1H-isochromen-1-one	−4.8
	Isorhamnetin 3–6 malonyl glycoside	−4.8
	Phellodendric acid	−4.7
	Metformin	−4.6
	Resveratrol	−8.0
	2-Chloro-n6-cyclopentyladenosine (gene agonist)	−7.3
*HCAR2*	Phaseic acid	−6.9
	Caffeic acid	−6.6
	4-hydoxycinnamic acid	−6.2
	Dodecanedioc acid	−5.6
	Sebaic acid	−4.9
	Citraconic acid	−4.8
	CNPD0447999	−4.7
	Pimelic acid	−4.7
	Sarmentose	−4.7
	Syndic acid	−4.7
	Glutaric acid	−4.6
	Metformin	−4.7
	Resveratrol	−6.5
	Butyric acid (gene agonist)	−3.4
*GABBR1*	Tetradecanedioc acid	−5.8
	Dodecanedioc acid	−5.7
	Methylisocitric acid	−5.6
	Quinic acid	−5.5
	xi-2,2,6-Trimethyl-1,4-cyclohexanedione	−5.4
	Sebaic acid	−5.1
	Pimelic acid	−5.0
	Glutaric acid	−4.7
	Metformin	−4.6
	Resveratrol	−6.4
	Baclofen (gene agonist)	−5.9

**Table 3 biology-12-01509-t003:** Thermodynamic components of identified secondary metabolites present in CS against target genes in cAMP pathway.

Energy Components (kcal/mol)					
Compound	ΔE_VdW_	ΔE_elec_	ΔG_gas_	ΔG_solv_	ΔG_bind_
*ADORA1*					
Cyperine	−34.29 ± 3.40	−12.89 ± 3.14	−47.18 ± 4.44	15.34 ± 2.60	−31.84 ± 3.68
Domesticoside	−42.99 ± 3.22	−19.59 ± 7.79	−62.58 ± 7.48	28.53 ± 6.08	−34.05 ± 3.72
Gallicynoic acid B	−47.88 ± 3.06	−18.32 ± 8.06	−66.20 ± 8.19	17.47 ± 4.65	−48.74 ± 4.86
Ginsenoyne e	−52.55 ± 3.32	−5.20 ± 2.36	−57.75 ± 4.24	9.87 ± 1.94	−34.05 ± 3.72
Quing hau sau	−40.15 ± 2.20	−5.63 ± 3.60	−45.78 ± 4.38	13.74 ± 3.58	−32.04 ± 2.56
Metformin	−2.78 ± 3.10	−93.28 ± 110.41	−96.05 ± 111.67	85.26 ± 104.62	−10.80 ± 7.76
Resveratrol	−6.80 ± 6.62	−8.12 ± 9.86	−14.92 ± 14.89	9.61 ± 9.82	−5.31 ± 5.62
*HCAR2*					
Caffeic acid	−17.94 ± 3.06	−37.51 ± 10.87	−55.45 ± 10.01	28.61 ± 8.60	−26.83 ± 3.60
Dodecanedioc acid	−36.58 ± 3.16	−31.29 ± 8.71	−67.87 ± 8.55	33.34 ± 6.29	−34.53 ± 4.21
4-hydoxycinnamic acid	−21.38 ± 2.04	−15.31 ± 9.12	−36.69 ± 8.70	22.11 ± 5.79	−14.50 ± 4.1
Phaseic acid	−29.88 ± 3.78	−8.14 ± 7.54	−34.80 ± 7.20	17.40 ± 6.55	−17.40 ± 3.90
Sebaic acid	−24.65 ± 3.89	−29.39 ± 15.43	−54.04 ± 13.34	32.12 ± 11.32	−21.92 ± 4.24
Metformin	−0.01 ± 0.15	107.28 ± 28.17	107.28 ± 28.15	−107.27 ± 28.14	0.01 ± 0.07
Resveratrol	−23.40 ± 5.18	−8.87 ± 4.24	−32.27 ± 5.84	15.97 ± 3.94	−16.31 ± 4.25
*GABBR1*					
Dodecanedioc acid	−31.61 ± 4.00	−43.03 ± 14.45	−74.64 ± 15.77	40.17 ± 11.38	−34.46 ± 5.56
Methylisocitric acid	−14.85 ± 3.36	−32.17 ± 14.21	−47.01 ± 13.67	29.01 ± 9.53	−18.00 ± 5.52
Quinic acid	−10.90 ± 4.99	−33.92 ± 20.44	−44.82 ± 22.61	31.78 ± 16.50	−13.04 ± 6.99
Tetradecanedioc acid	−28.26 ± 3.79	−45.28 ± 15.18	−73.54 ± 13.64	36.73 ± 9.51	−36.80 ± 5.25
Xi-2,2,6, trimethyl-1,4-cyclohexanedione	−15.51 ± 5.75	−5.03 ± 4.48	−20.55 ± 8.63	8.81 ± 4.35	−11.74 ± 5.15
Metformin	−2.53 ± 2.66	−273.87 ± 95.97	−276.40 ± 96.89	271.81 ± 93.74	−4.59 ± 4.68
Resveratrol	−28.78 ± 2.12	−11.87 ± 4.09	−40.65 ± 4.81	22.55 ± 2.82	−18.09 ± 2.87

∆E_vdW_: van der Waals energy; ∆E_elec_: electrostatic energy; ∆E_gas_: gas-phase free energy; ∆G_solv_ solvation free energy and ∆G_bind_: total binding free energy.

**Table 4 biology-12-01509-t004:** Post-molecular dynamics parameters of identified metabolites of CS against targets of cAMP pathway.

Compound	RMSD (Å)	RMSF (Å)	ROG (Å)	Number of H-bonds	SASA (Å)
*ADORA1*					
*ADORA1*	4.10 ± 0.60	1.93 ± 0.86	28.64 ± 0.39	173.53 ± 9.20	22,151.21 ± 602.21
Cyperine	5.93 ± 1.04	2.18 ± 1.16	29.04 ± 0.26	167.97 ± 9.25	22,745.86 ± 336.02
Domesticoside	4.81 ± 0.90	2.29 ± 1.43	28.55 ± 0.32	162.94 ± 9.61	22,760.33 ± 405.56
Gallicynoic acid B	6.71 ± 1.10	2.12 ± 1.59	28.39 ± 0.27	171.58 ± 9.69	21,996.14 ± 357.48
Ginsenoyne e	3.48 ± 0.53	2.12 ± 1.55	28.59 ± 0.27	170.93 ± 9.60	21,484.73± 426.66
Quing hau sau	4.06 ± 0.48	2.06 ± 0.98	28.51 ± 0.27	170.30 ± 9.68	22,366.55 ± 398.10
Metformin	6.26 ± 0.77	1.88 ± 0.86	28.39 ± 0.36	136.77 ± 8.90	17,546.87 ± 446.41
Resveratrol	4.10 ± 0.66	2.11 ± 1.02	29.16 ± 0.39	143.60 ± 8.90	18,462.50 ± 314.93
*HCAR2*					
*HCAR2*	9.64 ± 1.13	2.57 ± 2.39	24.24 ± 0.74	158.21 ± 10.58	20,865.81 ± 899.36
Caffeic acid	11.37 ± 1.67	2.25 ± 1.68	22.98 ± 0.63	166.40 ± 9.39	19,469.28 ± 645.20
Dodecanedioc acid	7.80 ± 0.80	2.21 ± 1.76	24.06 ± 0.33	160.98 ± 9.42	20,231.01 ± 515.88
4-hydoxycinnamic acid	9.46 ± 1.30	2.85 ± 2.29	23.70 ± 0.86	163.67 ± 9.43	20,516.00 ± 677.56
Phaseic acid	7.08 ± 0.53	2.07± 1.30	23.17 ± 0.27	158.70 ± 9.61	20,100.35 ± 631.16
Sebaic acid	9.73 ± 9.73	2.13 ± 1.55	23.20 ± 0.60	160.16 ± 9.14	20,546.94 ± 553.95
Metformin	9.54 ± 1.09	2.63 ± 1.91	24.70 ± 0.44	156.60 ± 9.51	20,894.76 ± 554.19
Resveratrol	9.10 ± 1.00	2.25 ± 1.96	23.79 ± 0.43	168.97 ± 10.01	20,736.27 ± 550.78
*GABBR1*					
*GABBR1*	1.97 ± 0.37	1.29 ± 0.50	23.11 ± 0.18	203.89 ± 9.56	17,313.73 ± 317.08
Dodecanedioc acid	2.21 ± 0.46	1.49± 0.57	23.49 ± 0.24	205.08 ± 9.77	17,372.46 ± 384.40
Methylisocitric acid	2.23 ± 0.39	1.27 ± 0.52	22.73 ± 0.18	205.83 ± 9.77	17,499.85 ± 327.27
Quinic acid	1.69 ± 0.28	1.34 ± 0.96	23.41 ± 0.19	211.12 ± 9. 11	17,117.23 ± 356.12
Tetradecanedioc acid	1.54 ± 0.24	1.25 ± 0.48	23.35 ± 0.17	211.77 ± 9.96	17,361.58 ± 314.41
Xi-2,2,6, trimethyl-1,4-cyclohexanedione	2.06 ± 0.38	1.58 ± 0.93	23.50 ± 0.25	207.06 ± 9.60	17,688.30 ± 388.06
Metformin	2.15 ± 0.47	1.63 ± 1.29	23.53 ± 0.22	207.00 ± 9.78	17,616.59 ± 385.58
Resveratrol	2.21 ± 0.35	1.22 ± 0.46	22.69 ± 0.14	205.87 ± 9.81	16,994.73 ± 319.80

RMSD: root mean square deviation, RMSF: root mean square fluctuation, ROG: radius of gyration: SASA: solvent accessible surface, H-bonds: hydrogen bonds.

**Table 5 biology-12-01509-t005:** The cDFT parameters of the top-hit compounds against target genes in the cAMP pathway.

					cDFT Parameters (eV)					
Ligands	LUMO	HUMO	EA	IE	EA	Hardness	Softness	EN	CP	GE
*ADORA1*										
Cyperine	−0.04	−5.75	5.71	0.04	5.75	2.85	0.35	2.89	−2.89	1.47
Domesticoside	−1.41	−6.29	4.88	1.41	6.29	2.44	0.41	3.85	−3.85	3.04
Gallicynoic acid B	−0.80	−6.63	5.83	0.80	6.63	2.91	0.34	3.72	−3.72	2.37
Ginsenoyne E	−2.58	−7.01	4.43	2.58	7.01	2.22	0.45	4.80	−4.80	5.19
Quing hau sau	−1.07	−7.13	6.06	1.07	7.13	3.03	0.33	4.10	−4.10	2.77
Metformin	−0.91	−6.17	5.26	0.91	6.17	2.63	0.38	3.54	−3.54	2.38
Reservatrol	−1.38	−5.44	4.06	1.38	5.44	2.03	0.49	3.41	−3.41	2.86
*HCAR2*										
Caffeic acid	−1.91	−6.05	4.14	1.91	6.05	2.07	0.48	3.98	−3.98	3.82
Dodecanedioc acid	0.45	−6.77	7.22	−0.45	6.77	3.61	0.28	3.16	−3.16	1.38
4-Hydroxycinnamic acid	−1.90	−6.17	4.27	1.90	6.17	2.14	0.47	4.03	−4.03	3.81
Phaseic acid	−2.30	−6.70	4.41	2.30	6.70	2.20	0.45	4.50	−4.50	4.59
Sebaic acid	−0.24	−7.65	7.41	0.24	7.65	3.70	0.27	3.94	−3.94	2.10
Metformin	−0.91	−6.17	5.26	0.91	6.17	2.63	0.38	3.54	−3.54	2.38
Reservatrol	−1.38	−5.44	4.06	1.38	5.44	2.03	0.49	3.41	−3.41	2.86
*GABBR1*										
Dodecanedioc acid	0.45	−6.77	7.22	−0.45	6.77	3.61	0.28	3.16	−3.16	1.38
Methylisocitric acid	−0.89	−7.54	6.65	0.89	7.54	3.33	0.30	4.21	−4.21	2.67
Quinic acid	−1.27	−6.78	5.51	1.27	6.78	2.76	0.36	4.03	−4.03	2.94
Tetradecanedioc acid	−0.22	−7.63	7.41	0.22	7.63	3.70	0.27	3.92	−3.92	2.08
Xi-2,2,6-Trimethyl-1,4-Cyclohexanedione	−1.25	−6.85	5.60	1.25	6.85	2.80	0.36	4.05	−4.05	2.93
Metformin	−0.91	−6.17	5.26	0.91	6.17	2.63	0.38	3.54	−3.54	2.38
Reservatrol	−1.38	−5.44	4.06	1.38	5.44	2.03	0.49	3.41	−3.41	2.86

cDFT: conceptual density functional theory; LUMO: lowest unoccupied molecular orbital; HOMO: highest occupied molecular orbital; EG: energy gap; IE: ionization energy; EA: electron effinity; EN: electronegativity; CP: chemical potentials; GE: global electropilicity.

## Data Availability

The data relating to the article are within the manuscript and in the Appendix A.

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
