# Peer review of "Waste to Medicine: Evidence from Computational Studies on the Modulatory Role of Corn Silk on the Therapeutic Targets Implicated in Type 2 Diabetes Mellitus"

_biology, 2023, doi:10.3390/biology12121509_

Round 1

Reviewer 1 Report

Comments and Suggestions for Authors

In this study, the authors used multiple approaches to explore the MOA behind the antidiabetic function of corn silk. The study is well-designed by comprehensively integrating solid evidence from multiple analyses like UP-LCMS, KEGG pathway enrichment, etc., identifying cAMP pathway and 3 candidate genes for T2DM therapy, suggesting the potential drug development via maintenance of normal glucose homeostasis and pancreatic β-cells function.

In general, the authors raised an interesting question and developed a solid narrative in this manuscript. There are only several minor suggestions:

Line 409, it might be interesting to have some explanation about the 5 non-interacting targets, what’s the function, and what might be the reason they’re not related.

Line 1445, it might be useful to add references for the ongoing research if there is any for the audience who is interested in the follow-up validation studies.

Author Response

Dear Reviewer,

We want to thank you for your valid comments arising from the review of our manuscript. Please find attached document for your magnanimous consideration. Thank you

Warm regards,

FO Balogun

Reviewer 2 Report

Comments and Suggestions for Authors

See attached file 

Comments on the Quality of English Language

minor revision 

Author Response

(The authors gave the same response as above.)

Reviewer 3 Report

Comments and Suggestions for Authors

Dear Author,

I hope this letter finds you well. I would like to praise you on the thorough research presented in your manuscript titled " Waste to Medicine: Evidence from computational studies on the modulatory role of corn silk on the therapeutic targets implicated in Type 2 diabetes mellitus"

I have carefully reviewed the manuscript and found it to be well-structured and supported by robust data. However, I believe that a few minor corrections and suggestions for improvement could enhance the clarity and impact of your findings. Kindly take into account the following points:

1.    In the molecular dynamics simulation, how was the stability of the complexes validated, and were the simulations run for a sufficient duration?

2.    Can you elaborate on how the identified compounds and their interactions contribute to the potential therapeutic effects of corn silk for T2DM?

3.    Rewrite the equations written in the line number 303.

4.    If possible use a clearer picture in Figure 5.

5.    Write a, b, c, and d embedded in the images of figure 8, like you did for figure 9 and other figures. 

Comments on the Quality of English Language

Although the manuscript is well written, there are some other minor grammatical errors and complex sentences that need to be addressed to enhance the clarity and readability of the paper. I recommend carefully reviewing the manuscript to ensure accurate grammar and punctuation, as well as considering breaking down some of the longer sentences to improve understanding.

Author Response

(The authors gave the same response as above.)
